# AUTOMATA : Gradient Based Data Subset Selection for Compute-Efficient Hyper-parameter Tuning

**Krishnateja Killamsetty[1],    Guttu Sai Abhishek[2],    Aakriti[2],    Ganesh Ramakrishnan[2]**
**Alexandre V. Evfimievski[3],    Lucian Popa[3],    Rishabh Iyer[1]**
[1] The University of Texas at Dallas, USA
[2]Indian Institute of Technology Bombay, India
[3] IBM Research, USA
{krishnateja.killamsetty, rishabh.iyer}@utdallas.edu
{gsaiabhishek, aakriti, ganesh}@cse.iitb.ac.in
{evfimi, lpopa}@us.ibm.com

## Abstract

Deep neural networks have seen great success in recent years; however, training a deep model is often challenging as its performance heavily depends on the hyper-parameters used. In addition, finding the optimal hyper-parameter configuration, even with state-of-the-art (SOTA) hyper-parameter optimization (HPO) algorithms, can be time-consuming, requiring multiple training runs over the entire dataset for different possible sets of hyper-parameters. Our central insight is that using an informative subset of the dataset for model training runs involved in hyper-parameter optimization, allows us to find the optimal hyper-parameter configuration significantly faster. In this work, we propose AUTOMATA, a gradient-based subset selection framework for hyper-parameter tuning. We empirically evaluate the effectiveness of AUTOMATA in hyper-parameter tuning through several experiments on real-world datasets in the text, vision, and tabular domains. Our experiments show that using gradient-based data subsets for hyper-parameter tuning achieves significantly faster turnaround times and speedups of **3×-30×** while achieving comparable performance to the hyper-parameters found using the entire dataset.

## 1   Introduction

In recent years, deep learning systems have found great success in a wide range of tasks, such as object recognition [15], speech recognition [17], and machine translation [1], making people's lives easier on a daily basis. However, in the quest for near-human performance, more complex and deeper machine learning models trained on increasingly large datasets are being used at the expense of substantial computational costs. Furthermore, deep learning is associated with a significantly large number of hyper-parameters such as the learning algorithm, batch size, learning rate, and model configuration parameters (*e.g.*, depth, number of hidden layers, *etc.*) that need to be tuned. Hence, running extensive hyper-parameter tuning and auto-ml pipelines is becoming increasingly necessary to achieve state-of-the-art models. However, tuning the hyper-parameters requires multiple training runs over the entire datasets (which are significantly large nowadays), resulting in staggering compute costs, running times, and, more importantly, CO2 emissions.

To give an idea of staggering compute costs, we consider an image classification task on a relatively simple CIFAR-10 dataset where a single training run using a relatively simple model class of Residual Networks [16] for 300 epochs on a V100 GPU takes around 4 hours. If we perform 1000 training runs (which is not uncommon today) naively using grid search for hyper-parameter tuning, it will take 4000 GPU hours. The resulting CO2 emissions would be between 440 to 1000 kg of CO2

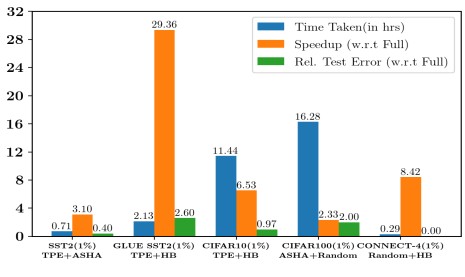 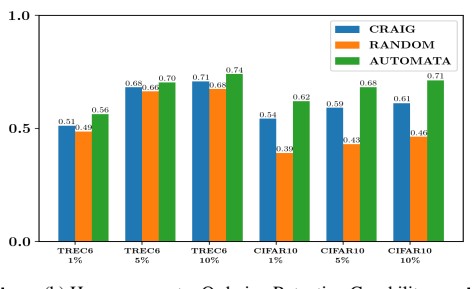

(a) Hyper-parameter Tuning Performance      (b) Hyper-parameter Ordering Retention Capability

**Figure 1:** Sub-figure Figure 1a shows performance summary of AUTOMATA with speedups, relative test errors, and tuning times on SST2, glue-SST2, CIFAR10, CIFAR100, and CONNECT-4 datasets. We observe that AUTOMATA achieves speedups(and similar energy savings) of 10x - 30x with around 2% performance loss using Hyperband as a scheduler. Similarly, even when using a more efficient ASHA scheduler, AUTOMATA achieves a speedup of around 2x-3x with a performance loss of 0%-2%. Sub-figure Figure 1b shows the Spearman ranking Correlation values of hyper-parameter ordering returned by AUTOMATA and hyper-parameter ordering returned by full training using Grid Search. Spearman ranking correlation values suggest that AUTOMATA was able to retain hyper-parameter ordering better than other subset selection baselines even when using small data subsets for individual model training.

emitted[1], which is equivalent to 1100 to 2500 miles of car travel in the US. Similarly, the costs of training state-of-the-art NLP models and vision models on larger datasets like ImageNet are even more staggering [48][2].

Naive hyper-parameter tuning methods like grid search [2] often fail to scale up with the dimensionality of the search space and are computationally expensive. Hence, more efficient and sophisticated Bayesian optimization methods [4, 19, 5, 46, 27] have dominated the field of hyper-parameter optimization in recent years. Bayesian optimization methods aim to identify good hyper-parameter configurations quickly by building a posterior distribution over the search space and by adaptively selecting configurations based on the probability distribution. More recent methods [49, 50, 10, 27] try to speed up configuration evaluations for efficient hyper-parameter search; these approaches speed up the configuration evaluation by adaptively allocating more resources to promising hyper-parameter configurations while eliminating poor ones quickly.

Recent works like SHA [20], Hyperband [32], ASHA [34] use aggressive early-stopping strategies to stop not-so-promising configurations quickly while allocating more resources to the promising ones. Generally, these resources can be the size of the training set, number of gradient descent iterations, training time, etc. Nickson et al. [40], Krueger et al. [29] try to quickly evaluate a configuration's performance on a large dataset by evaluating the training runs on small, random subsets and they empirically show that small data subsets could suffice to estimate a configuration's quality. Similarly, past works [40, 29] show that very small data subsets can be effectively used to find the best hyper-parameters quickly. However, all these approaches have naively used random training data subsets and did not place much focus on selecting informative subsets instead. Our central insight is that using small informative data subsets allows us to find good hyper-parameter configurations more effectively than random data subsets.

An earlier work called GRADMATCH [23] demonstrated the effectiveness of gradient-based data subset selection for selecting informative data subsets by using it for efficient model training in a supervised learning setting. In this work, we empirically investigate the advantage of using gradient-based subset selection algorithms, mainly GRADMATCH, for hyper-parameter tuning compared to using random subsets and the full dataset for hyper-parameter tuning. Fundamentally, we use GRADMATCH to select data subsets that can be used for efficiently tuning the hyper-parameters. To this end, we propose AUTOMATA, an efficient hyper-parameter tuning framework that combines existing hyper-parameter search and scheduling algorithms with intelligent subset selection. Further, our experimental results illustrate that very small subsets (1%, 5% subsets) can be used for effective hyper-parameter tuning because we are primarily concerned with retaining the ordering of hyper-parameters rather than the final accuracy, thereby enabling us to achieve better speedups and energy efficiency (and more significant CO2 emissions reduction).

---

[1] https://mlco2.github.io/impact/#compute
[2] https://tinyurl.com/a66fexc7

## 1.1 Related Work

**Hyper-parameter Tuning Approaches:** A number of algorithms have been proposed for hyper-parameter tuning including grid search[3], Bayesian algorithms [3], random search [43], etc. Furthermore, a number of scalable toolkits and platforms for hyper-parameter tuning exist like Ray-tune [36], H2O automl [31], etc. See [45, 54] for a survey of current approaches and also tricks for hyper-parameter tuning for deep models. The biggest challenges of existing hyper-parameter tuning approaches are a) the large hyper-parameter search space and b) the increased training times of training models. Recently, Li *et. al.* [33] have proposed an efficient approach for parallelizing hyper-parameter tuning using Asynchronous Successive Halving Algorithm (ASHA). AUTOMATA is complementary to such approaches and can be be combined effectively with them.

**Data Subset Selection Approaches:** Several recent papers have used submodular functions[4] for data subset selection towards various applications like speech recognition [53, 52], machine translation [26] and computer vision [21]. Other common approaches for subset selection include the usage of coresets. Coresets are weighted subsets of the data, which approximate certain desirable characteristics of the full data (*e.g.*, the loss function) [12]. Coreset algorithms have been used for several problems including $k$-means and $k$-median clustering [14], SVMs [8] and Bayesian inference [6]. Recent coreset selection-based methods [38, 24, 23, 25] have shown great promise for efficient and robust training of deep models. CRAIG [38] tries to select a coreset summary of the training data that estimate the full training gradient closely. Whereas GLISTER [24] poses the coreset selection problem as a discrete-continuous bilevel optimization problem that minimizes the validation set loss. Similarly, RETRIEVE [25] also uses a discrete bilevel coreset selection problem to select unlabeled data subsets for efficient semi-supervised learning. Another approach GRAD-MATCH [23] selects coreset summary that approximately matches the full training loss gradient using orthogonal matching pursuit.

The contributions of our work can be summarized as follows: **AUTOMATA Framework:** We propose AUTOMATA a framework that combines intelligent gradient based subset selection with hyper-parameter search and scheduling algorithms to enable faster hyper-parameter tuning. Our work empirically studies the role of intelligent data subset selection for hyper-parameter tuning. In particular, we seek to answer the following question: *Is it possible to use small informative data subsets between 1% to 30% for faster configuration evaluations in hyper-parameter tuning, thereby enabling faster tuning times while maintaining comparable accuracy to tuning hyper-parameters on the full dataset?* **Importance of Intelligent Subset Selection:** Our empirical results in Figure 1b demonstrate that GRADMATCH [23] adopted by AUTOMATA preserves the original hyperparameter ordering better than other baseline subset selection strategies including RANDOM and CRAIG. In addition, gradient-based subset selection strategies such as GRADMATCH and CRAIG preserve the original hyper-parameter ordering better compared to RANDOM, thereby showcasing the importance of intelligent subset selection approaches. **Effectiveness of AUTOMATA :** We empirically demonstrate the effectiveness of AUTOMATA framework used in conjunction with existing hyper-parameter search algorithms like TPE, Random Search, and hyper-parameter scheduling algorithms like Hyperband, and ASHA through a set of extensive experiments on multiple real-world datasets. We give a summary of the speedup vs. relative performance achieved by AUTOMATA compared to full data training in Figure 1a. More specifically, AUTOMATA achieves a speedup of 3x - 30x with minimal performance loss for hyper-parameter tuning. Further, in Section 3, we show that the gradient-based subset selection approach of AUTOMATA outperforms the previously considered random subset selection for hyper-parameter tuning.

## 2 AUTOMATA Framework

In this section, we present AUTOMATA a hyper-parameter tuning framework, and discuss its different components shown in Figure 2. The AUTOMATA framework consists of three components: a hyper-parameter search algorithm that identifies which configuration sets need to be evaluated, a gradient-based subset selection algorithm for training and evaluating each configuration efficiently,

---

[3]https://tinyurl.com/3hb2hans
[4]Let $V = \{1, 2, \cdots, n\}$ denote a ground set of items. A set function $f : 2^V \to \mathbf{R}$ is a submodular [13] if it satisfies the diminishing returns property: for subsets $S \subseteq T \subseteq V$ and $j \in V \backslash T, f(j|S) \triangleq f(S \cup j) - f(S) \geq f(j|T)$.

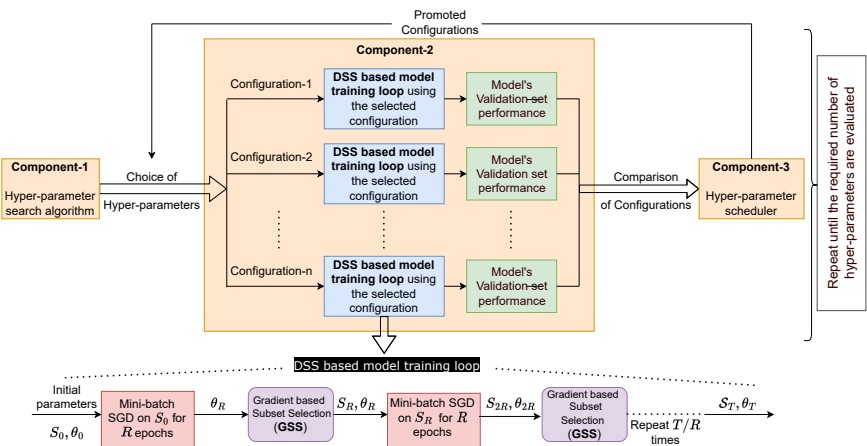

Figure 2: Pipeline figure of AUTOMATA including hyper-parameter search, subset based configuration evaluation (where models are trained on subsets of data), and hyper-parameter scheduler.

and a hyper-parameter scheduling algorithm that provides early stopping by eliminating the poor configurations quickly. With AUTOMATA framework, one can use any of the existing hyper-parameter search and hyper-parameter scheduling algorithms and still achieve significant speedups with minimal performance degradation due to faster configuration evaluation using gradient-based subset training.

**Notation** Denote by $\mathbb{H}$, the set of configurations selected by the hyper-parameter search algorithm. Let $\mathcal{D} = \{(x_i, y_i)\}_{i=1}^{N}$, denote the set of training examples, and $\mathcal{V} = \{(x_j, y_j)\}_{j=1}^{M}$ the validation set. Let $\theta_i$ denote the classifier model parameters trained using the configuration $i \in \mathbb{H}$. Let $\mathcal{S}_i$ be the subset used for training the $i^{th}$ configuration model $\theta_i$ and $w_i$ be its associated weight vector i.e., each data sample in the subset has an associated weight that is used for computing the weighted loss. We superscript the changing variables like model parameters $\theta$, subset $\mathcal{S}$ with the timestep $t$ to denote their specific values at that timestep. Next, denote by $L_T^j(\theta_i) = L_T(x_j, y_j, \theta_i)$, the training loss of the $j^{th}$ data sample in the dataset for $i^{th}$ classifier model, and let $L_T(\theta_i) = \sum_{k \in \mathcal{D}} L_T^k(\theta_i)$ be the loss over the entire training set for $i^{th}$ configuration model. Let, $L_T^j(\mathcal{S}, \theta_i) = \sum_{k \in \mathcal{X}} L_T(x_k, y_k, \theta_i)$ be the loss on a subset $\mathcal{S} \subseteq \mathcal{V}$ of the training examples at timestep $j$. Let the validation loss be denoted by $L_V$.

**Component-1: Hyper-parameter Search Algorithm -** Given a hyper-parameter search space, hyper-parameter search algorithms provide a set of configurations that need to be evaluated. A naive way of performing the hyper-parameter search is Grid-Search, which defines the search space as a grid and exhaustively evaluates each grid configuration. However, Grid-Search is a time-consuming process, meaning that thousands to millions of configurations would need to be evaluated if the hyper-parameter space is large. In order to find optimal hyper-parameter settings quickly, Bayesian optimization-based hyper-parameter search algorithms have been developed. To investigate the effectiveness of AUTOMATA across the spectrum of search algorithms, we used the Random Search method and the Bayesian optimization-based TPE method as representative hyper-parameter search algorithms. We provide more details on Random Search and TPE in Appendix E.

**Component-2: Subset based Configuration Evaluation -** Earlier, we discussed how a hyper-parameter search algorithm presents a set of potential hyper-parameter configurations that need to be evaluated when tuning hyper-parameters. Every time a configuration needs to be evaluated, prior work trained the model on the entire dataset until the resource allocated by the hyper-parameter scheduler is exhausted. Rather than using the entire dataset for training, we propose using subsets of informative data selected based on gradients instead. As a result, given any hyper-parameter search algorithm, we can use the data subset selection to speed up each training epoch by a significant factor *(say 10x)*, thus improving the overall turnaround time of the hyper-parameter tuning. We use GRADMATCH [23], a gradient based subset selection strategy in AUTOMATA because GRADMATCH is able to preserve the original ordering of hyper-parameters better than RANDOM and other baselines even when using small subset sizes as shown in Figure 1b. Hence, the critical advantage of AUTOMATA is that we can achieve speedups while still retaining the hyper-parameter tuning algorithm's performance in

finding the best hyper-parameters. We provide more details on hyper-parameter ordering retention experiments in Section 3. The fundamental feature of AUTOMATA is that the subset selected by AUTOMATA changes adaptively over time, based on the classifier model training. Thus, instead of selecting a common subset among all configurations, AUTOMATA selects the subset that best suits each configuration. We present the gradient-based subset selection process of AUTOMATA below.

***Gradient Based Subset Selection (GSS):*** AUTOMATA adopts GRADMATCH [23], a gradient based subset-selection strategy, to select a subset $\mathcal{S}$ and its associated weight vector $\boldsymbol{w}$ such that the weighted subset loss gradient best approximates the entire training loss gradient. The subset selection of AUTOMATA for $i^{th}$ configuration at time step $t$ is as follows:

$$\boldsymbol{w}_i^t, \mathcal{S}_i^t = \underset{\boldsymbol{w}_i^t, \mathcal{S}_i^t : |\mathcal{S}_i^t| \leq k, \boldsymbol{w}_i^t \geq 0}{\operatorname{argmin}} \| \sum_{l \in \mathcal{S}_i^t} \boldsymbol{w}_{il}^t \nabla_\theta L_T^l(\theta_i^t) - \nabla_\theta L_T(\theta_i^t) \| + \lambda \left\| \boldsymbol{w}_i^t \right\|^2 \tag{1}$$

The additional regularization term prevents the assignment of very large weight values to data samples, thereby reducing the possibility of overfitting on a few data samples. Killamsetty et al. [23] proved that the optimization problem given in Equation (1) is approximately submodular. Therefore, the above optimization problem can be solved using greedy algorithms with approximation guarantees [9, 37]. Therefore, like Killamsetty et al. [23], we also use a greedy algorithm called orthogonal matching pursuit (OMP) to solve the above optimization problem. The goal of AUTOMATA is to accelerate the hyper-parameter tuning algorithm while preserving its original performance. Efficiency is an essential factor that AUTOMATA considers even when selecting subsets. Due to this, we employ a faster per-batch subset selection introduced in the work [23] in our experiments, which is described in the following section.

***Per-Batch Subset Selection:*** Instead of selecting a subset of data points, one selects a subset of mini-batches by matching the weighted sum of mini-batch training gradients to the full training loss gradients. We visualize the difference between per-sample and per-batch subset selection in Figure 5 of the Appendix. Therefore, one will have a subset of selected mini-batches and the associated mini-batch weights. One trains the model on the selected mini-batches by performing mini-batch gradient descent using the weighted mini-batch loss. Let us denote the batch size as $B$, and the total number of mini-batches as $b_N = \frac{N}{B}$, and the training set of mini-batches as $\mathcal{D}_\mathcal{B}$. Let us denote the number of mini-batches that needs to be selected as $b_k = \frac{k}{B}$. Let us denote the subset of mini-batches that needs to be selected as $\mathcal{S}_{\mathcal{B}i}$ and denote the weights associated with mini-batches as $\boldsymbol{w}_{\mathcal{B}i} = \{\boldsymbol{w}_{\mathcal{B}i1}, \boldsymbol{w}_{\mathcal{B}i2} \cdots \boldsymbol{w}_{\mathcal{B}ik}\}$ for the $i^{th}$ model configuration. Let us denote the mini-batch gradients as $\nabla_\theta L_T^{\mathcal{B}_1}(\theta_i), \cdots, \nabla_\theta L_T^{\mathcal{B}_{b_N}}(\theta_i)$ be the mini-batch gradients for the $i^{th}$ model configuration. Let us denote $L_T^{\mathcal{B}}(\theta_i) = \sum_{k \in [1, b_N]} L_T^{\mathcal{B}_k}(\theta_i)$ be the loss over the entire training set. The subset selection problem of mini-batches at time step $t$ can be written as follows:

$$\boldsymbol{w}_{\mathcal{B}i}^t, \mathcal{S}_{\mathcal{B}i}^t = \underset{\boldsymbol{w}_{\mathcal{B}i}^t, \mathcal{S}_{\mathcal{B}i}^t : |\mathcal{S}_{\mathcal{B}i}^t| \leq b_k, \boldsymbol{w}_{\mathcal{B}i}^t \geq 0}{\operatorname{argmin}} \| \sum_{l \in \mathcal{S}_{\mathcal{B}i}^t} \boldsymbol{w}_{\mathcal{B}l}^t \nabla_\theta L_T^{\mathcal{B}_l}(\theta_i^t) - \nabla_\theta L_T^{\mathcal{B}}(\theta_i^t) \| + \lambda \left\| \boldsymbol{w}_{\mathcal{B}i}^t \right\|^2 \tag{2}$$

In the per-batch version, because the number of samples required for selection is $b_k$ is less than $k$, the number of greedy iterations required for data subset selection in OMP is reduced, resulting in a speedup of $B\times$. A critical trade-off in using larger batch sizes is that in order to get better speedups, we must also sacrifice data subset selection performance. Therefore, it is recommended to use smaller batch sizes for subset selection to get a optimal trade-off between speedups and performance. A solution to the batch size dependency is to use a smaller batch size only for subset selection and combine the smaller batches to produce larger batches for training. In order to demonstrate that the effectiveness of AUTOMATA is not dependent on the size of the training batch, in our experiments on Images, we use a fixed batch size (instead of as a hyper-parameter) of $B = 20$, and on text datasets, we use the batch size as a hyper-parameter with $B \in [16, 32, 64]$. Apart from per-batch selection, we use model warm-starting to get more informative data subsets. Further, in our experiments, we use a regularization coefficient of $\lambda = 0$ after analyzing the performance of AUTOMATA with different values of $\lambda$. We present the ablation study on $\lambda$ in Appendix G.3.1. We give more details on warm-starting below.

***Warm-starting data selection:*** We warm-start each configuration model by training on the entire training dataset for a few epochs similar to [23]. The warm-starting process enables the model to

have informative loss gradients used for subset selection. To be more specific, the classifier model is trained on the entire training data for $T_w = \frac{\kappa T k}{N}$ epochs, where $k$ is the coreset size, $T$ is the total number of epochs, $\kappa$ is the fraction of warm start, and $N$ is the size of the training dataset. We use a $\kappa$ value of 0 (*i.e.*, no warm start) for experiments using Hyperband as scheduling algorithm, and a $\kappa$ value of 0.35 for experiments using ASHA. We present the ablation studies for $\kappa$ in Appendix G.3.2.

**Component-3: Hyper-parameter Scheduling Algorithm -** Hyper-parameter scheduling algorithms improve the overall efficiency of the hyper-parameter tuning by terminating some of the poor configurations runs early. In our experiments, we consider Hyperband [32], and ASHA [34], which are extensions of the Sequential Halving algorithm (SHA) [20] that uses aggressive early stopping to terminate poor configuration runs and allocates an increasingly exponential amount of resources to the better performing configurations. SHA starts with $n$ number of initial configurations, each assigned with a minimum resource amount $r$. The SHA algorithm uses a reduction factor $\eta$ to reduce the number of configurations in each round by selecting the top ${\frac{1}{\eta}}^{th}$ fraction of configurations while also increasing the resources allocated to these configurations by $\eta$ times each round. We discuss Hyperband and ASHA and the issues within SHA that each of them addresses in more detail in Appendix F. Detailed pseudocode of the AUTOMATA algorithm is provided in Appendix C due to space constraints in the main paper. We use the popular deep learning framework [41] for implementation of AUTOMATA framework, Ray-tune[36] for hyper-parameter search and scheduling algorithms, and CORDS [22] for subset selection strategies.

## 3 Experiments

In this section, we present the effectiveness and the efficiency of AUTOMATA framework for hyper-parameter tuning by evaluating AUTOMATA on datasets spanning text, image, and tabular domains. Further, to assess AUTOMATAś effectiveness across the spectrum of existing hyper-parameter search and scheduling algorithms, we conduct experiments using combinations of different search and scheduling algorithms. As discussed earlier, we employ Random Search [43], TPE [4] as representative hyper-parameter search algorithms, and Hyperband [32], ASHA [34] as representative hyper-parameter scheduling algorithms. It should be noted that the combination of TPE and HyperBand is not the same as the BOHB algorithm [11]. In BOHB [11], a multivariate kernel density estimator (KDE) was used, whereas the TPE [4] used a hierarchy of one-dimensional KDEs. However, we believe the takeaways would remain the same even with other approaches. We repeat each experiment five times on the text and tabular datasets, thrice on the image datasets, and report the mean accuracy and speedups in the plots. Below, we provide further details on datasets, baselines, models, and the hyper-parameter search space used for experiments.

**Baselines:** Our experiments aim to demonstrate the consistency and efficiency of AUTOMATA more specifically, the effectiveness of AUTOMATA's gradient-based subset selection (GSS) for hyper-parameter tuning. As baselines, we replace the GSS subset selection strategy in AUTOMATA with different subset selection strategies, namely RANDOM (randomly sample a same sized subset as AUTOMATA from the training data), CRAIG [38] (a gradient-based subset selection proposed for efficient supervised learning), and FULL (using the entire training data for model training during configuration evaluation). For ease of notation, we refer to baselines by the names of corresponding subset selection strategies. Note that by CRAIG baseline, we mean the faster per-batch version of CRAIG [38] for subset selection shown [23] to be more efficient than the original. In addition, for all methods, we do not use any warm-start for experiments with Hyperband and use a warm start of $\kappa = 0.35$ for experiments with ASHA. We give more details on the reason for using warm-start with ASHA and no warm-start with Hyperband in Appendix G.3.2. We perform experiments with different subset size fractions of 1%, 5%, 10%, and 30%. In our experiments, we compare our approach's accuracy and efficiency (time/energy) with Full training, Per Batch CRAIG, and Random selection.

**Datasets, Model Architecture, and Experimental Setup:** To demonstrate the effectiveness of AUTOMATA for hyper-parameter tuning, we performed experiments on datasets spanning text, image and tabular domains. Text datasets include SST2 [47], SST5 [47], glue-SST2 [51], and TREC6 [35, 18]. Image datasets include CIFAR10 [28], CIFAR100 [28], and Street View House Numbers (SVHN) [39]. Tabular datasets include DNA, SATIMAGE, LETTER, and CONNECT-4 from **LIBSVM** (a library for Support Vector Machines (SVMs)) [7]. We give more details on dataset sizes and splits in Appendix G.2. For the Text datasets, we use the LSTM model (from PyTorch)

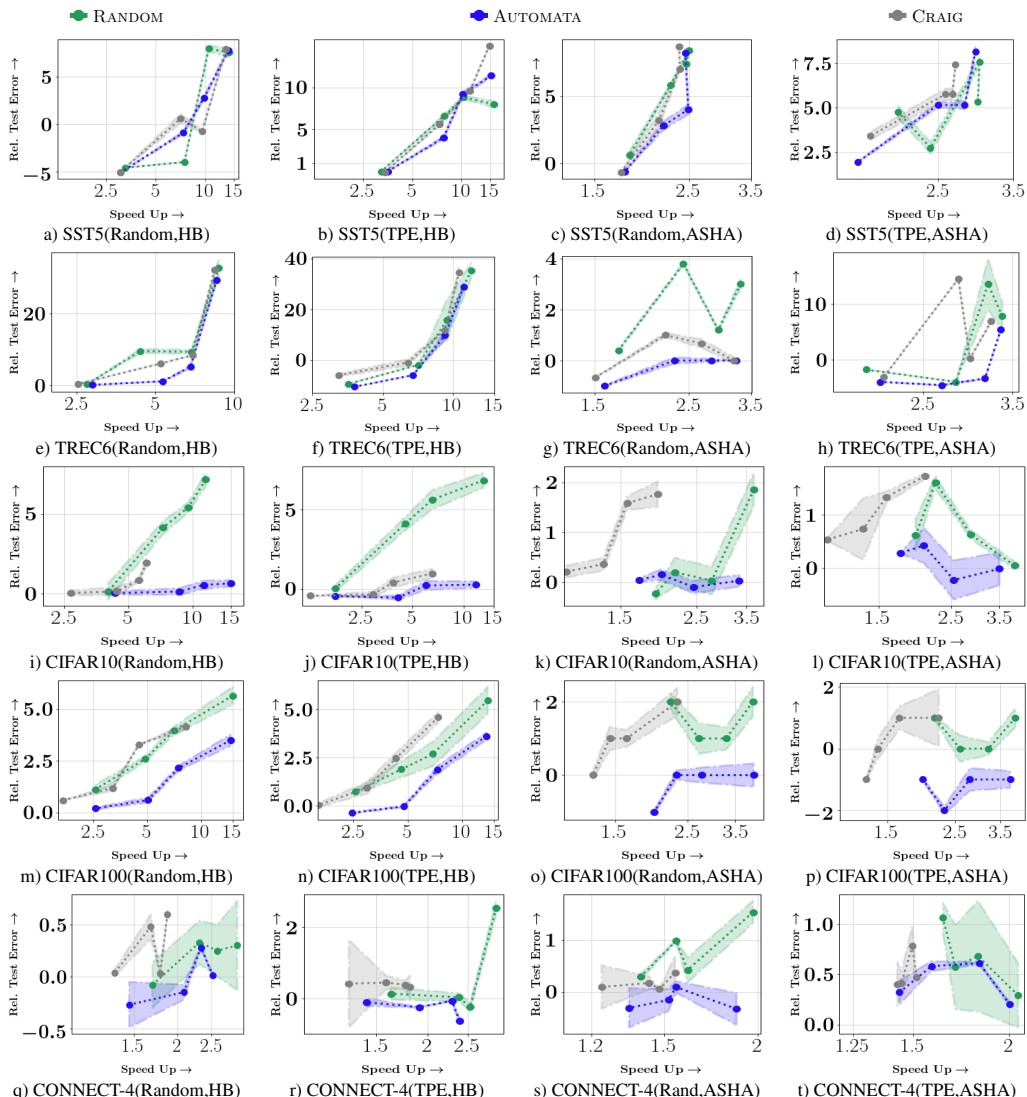

Figure 3: Comparison of performance of AUTOMATA with baselines(RANDOM, CRAIG, FULL) for Hyper-parameter tuning. In sub-figures (a-t), we present speedup *vs.* relative test error (in %), compared to Full data tuning for different methods. A **relative test error (%)** is the difference between a test error obtained through subset selection based tuning and a test error obtained through full data tuning. On each scatter plot, smaller subsets appear on the right, and larger ones appear on the left. Results are shown for (a-d) SST5, (e-h) TREC6, (i-l) CIFAR10, (m-p) CIFAR100, and (q-t) CONNECT-4 datasets with different combinations of hyper-parameter search and scheduling algorithms. *The scatter plots show that* AUTOMATA *achieves the best speedup-accuracy tradeoff in almost every case (**bottom-right corner of each plot indicates the best speedup-accuracy tradeoff region**).*

with trainable GloVe [42] embeddings of 300 dimension as input. For Image datasets, we use the ResNet18 [16] and ResNet50 [16] models. For Tabular datasets, we use a multi-layer perceptron with 2 hidden layers. Once the best hyper-parameter configuration is found, we perform one more *final training* of the model using the best configuration on the entire dataset and report the achieved test accuracy. We use *final training* for all methods except FULL since the models trained on small data subsets (especially with small subset fractions of 1%, 5%) during tuning do not achieve high test accuracies. We also include the final training times while calculating the tuning times for a more fair comparison[5] For text datasets, we train the LSTM model for 20 epochs while choosing subsets (except for FULL) every 5 epochs. The hyper-parameter space includes learning rate, hidden size & number of layers of LSTM, batch size of training. Some experiments (with TPE as the search algorithm) use 27 configurations in the hyper-parameter space, while others use 54. More details on hyper-parameter search space for text datasets are given in Appendix G.4.1. For image datasets, we train the ResNet [16] model for 300 epochs while choosing subsets (except for FULL) every 20 epochs i.e., $R = 20$. We present an ablation study analyzing the effect of epoch interval ($R$) on the performance of AUTOMATA in Appendix G.3.3. Based on the results, we observe that for vision experiments, using a $R$ value of 20 gives best efficiency vs performance tradeoff. We use a Stochastic Gradient Descent (SGD) optimizer with momentum set to 0.9 and weight decay factor set to 0.0005. The hyper-parameter search space consists of a choice between the Momentum method and Nesterov Accelerated Gradient method, choice of learning rate scheduler and their corresponding parameters, and four different group-wise learning rates. We use 27 configurations in the hyper-parameter space for Image datasets. More details on hyper-parameter search space for image datasets are given in Appendix G.4.2. For tabular datasets, we train a multi-layer perceptron with 2 hidden layers for 200 epochs while choosing subsets every 10 epochs. The hyper-parameter search space consists of a choice between the SGD optimizer or Adam optimizer, choice of learning rate, choice of learning rate scheduler, the sizes of the two hidden layers and batch size for training. We use 27 configurations in the hyper-parameter space for Tabular datasets. More details on hyper-parameter search space for tabular datasets are provided in Appendix G.4.3.

**Hyper-parameter Ordering Retention Experiments:** We evaluate the effectiveness of different subset selection strategies in preserving original hyper-parameter ordering by comparing the ordering obtained using AUTOMATA's gradient-based subset selection (GSS), RANDOM, CRAIG strategies with that obtained using full data. Intuitively, we want to analyze whether the original hyper-parameter ordering is preserved even when using small subsets for model training. To examine this, we experiment on the CIFAR10 [28], Trec6 [35, 18] datasets using a ResNet18 [16] model and LSTM model respectively. Hyper-parameter search for CIFAR10 dataset includes a grid search over 144 configurations of four group-wise learning rates of ResNet18 model, optimizer, and training batch size. Hyper-parameter search for TREC6 dataset includes a grid search over 108 configurations of learning rate, optimizer, LSTM hidden size, training batch size, and number of final fully connected layers. Figure 1b shows the Spearman rank correlation values between the hyper-parameter ordering obtained using 1%, 5%, and 10% subsets selected by RANDOM, CRAIG, and AUTOMATA's gradient-based subset selection (GSS) and Full data hyper-parameter ordering on CIFAR10 [28] and TREC6 [35, 18] datasets. Figure 1b demonstrates that GSS is more effective than RANDOM and CRAIG in preserving hyper-parameter ordering even when using small subsets.

**Hyper-parameter Tuning Results:** Results comparing the accuracy vs. efficiency tradeoff of different subset selection strategies for hyper-parameter tuning are shown in Figure 3. Performance is compared for different sizes of subsets of training data: 1%, 5%, 10%, and 30% along with four possible combinations of search algorithm (Random or TPE) and scheduling algorithm (ASHA or Hyperband). *Text datasets results:* Sub-figures(3a, 3b, 3c, 3d) show the plots of relative test error *vs.* speed ups, both *w.r.t* full data tuning for SST5 dataset with different combinations of search and scheduling methods. Similarly, in sub-figures(3e, 3f, 3g, 3h) we present the plots of relative test error *vs.* speed ups for TREC6 dataset. From the results, we observe that AUTOMATA achieves best speed up *vs.* accuracy tradeoff and consistently gives better performance even with small subset sizes unlike other baselines like RANDOM, CRAIG. In particular, AUTOMATA achieves a speedup of $9.8\times$ and $7.35\times$ with a performance loss of 2.8% and a performance gain of 0.9% respectively on the SST5 dataset with TPE and Hyperband. Additionally, AUTOMATA achieves a speedup of around $3.15\times$, $2.68\times$ with a performance gain of 3.4%, 4.6% respectively for the TREC6 dataset with TPE and ASHA.

---

[5]Note that with a 30% subset, final training is not required as the models trained with 30% subsets achieve similar accuracy to full data training. However, for the sake of consistency, we use *final training* with 30% subsets as well.

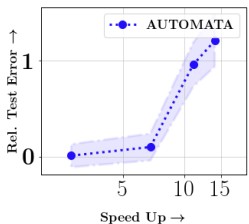

Figure 4: Performance of AU-TOMATA on CIFAR10 dataset with 110 search configurations

***Image datasets results:*** Sub-figures(3i, 3j, 3k, 3l) show the plots of relative test error *vs.* speed ups, both *w.r.t* full data tuning for CIFAR10 dataset with different combinations of search and scheduling methods. Similarly, sub-figures (3m, 3n, 3o, 3p) show the plots of relative test error *vs.* speed ups on CIFAR100. The results show that AUTOMATA achieves the best speed up *vs.* accuracy tradeoff consistently compared to other baselines. More specifically, AUTOMATA achieves a speedup of around $15\times$, $8.7\times$ with a performance loss of $0.65\%$, $0.14\%$ respectively on the CIFAR10 dataset with Random and Hyperband. Further, AUTOMATA achieves a speedup of around $3.7\times$, $2.3\times$ with a performance gain of $1\%$, $2\%$ for CIFAR100 dataset with TPE and ASHA. In Figure 4, we show the effectiveness of the AUTOMATA with large configuration space by repeating the experiment on the CIFAR10 dataset with 110 configurations using Random as the search algorithm and Hyperband as the scheduler.

***Tabular datasets results:*** Sub-figures(3q, 3r, 3s, 3t) show the plots of relative test error *vs.* speed ups for the CONNECT-4 dataset. In practice, larger subset sizes should lead to better performance. However, the plots(3q, 3r, 3s, 3t) do not depict the same picture. This is due to the inherent class imbalance within the CONNECT-4 dataset. In this case, smaller subset sizes with gradient based subset selection mitigated the effects of class imbalance and enabled better model performance by selecting a similar proportion of samples from each class. Furthermore, using larger subset sizes caused gradient-based subset selection to select more samples from overrepresentative classes, thus reducing the final model's accuracy. Finally, AUTOMATA consistently achieved better speedup *vs.* accuracy tradeoff compared to other baselines on CONNECT-4 as well.

Despite the fact that CRAIG and GSS have similar optimization problems, the performance of CRAIG is poor compared to AUTOMATA since CRAIG suboptimally optimizes an upper bound of the gradient error term. In addition, CRAIG is slower from a computational complexity standpoint since it requires the construction of a similarity kernel. Owing to space constraints, we provide additional results showing the accuracy *vs.* efficiency tradeoff on additional text, image, and tabular datasets in the Appendix G.5. It is important to note that AUTOMATA obtains better speedups when used for hyper-parameter tuning on larger datasets and larger models (in terms of parameters). Apart from the speedups achieved by AUTOMATA we show in Appendix G.6 that it also achieves similar reductions of energy consumption and CO2 emissions, thereby making it more environmentally friendly. AUTOMATA can be used in conjunction with any hyper-parameter search algorithm and scheduler. However, It should be noted that the speedups obtained using AUTOMATA are heavily dependent upon the scheduler. Some schedulers, such as ASHA, are highly efficient and can effectively discard poor hyper-parameter configurations early, thus reducing the advantages caused by using data subsets. Therefore, the speedup achieved by AUTOMATA is approximately 3x when using ASHA as a scheduler and around 10x-15x when using Hyperband. Nevertheless, we observe that using AUTOMATA still reduces tuning time and thus power consumption.

## 4   Conclusion, Limitations, and Broader Impact

We introduce AUTOMATA an efficient hyper-parameter tuning framework that uses intelligent subset selection for model training for faster configuration evaluations. Further, we perform extensive experiments showing the effectiveness of AUTOMATA for Hyper-parameter tuning. In particular, it achieves speedups of around $10\times$ - $15\times$ using Hyperband as scheduler and speedups of around $3\times$ even with a more efficient ASHA scheduler. AUTOMATA significantly decreases CO2 emissions and energy-efficient, reducing the environmental impact of hyper-parameter tuning on society at large. We hope that the AUTOMATA as framework will encourage community to consider subset selection approaches for faster hyper-parameter tuning, helping us move closer to the goal of Green AI [44]. Research like this will help substantially reduce the cost of training large models and AutoML, thereby help democratize machine learning among smaller companies, individuals and academic groups. One of the limitations of AUTOMATA is that in scenarios in which no performance loss is desired, we do not know the minimum subset size that gives the best speed up and, therefore, need to use larger subset sizes such as 10%, 30%. In the future, we consider adapting the subset size based on model performance for each configuration to remove the dependency on subset size.

## Acknowledgments and Disclosure of Funding

Rishabh Iyer and Krishnateja Killamsetty would like to acknowledge support from NSF Grant Number IIS-2106937, a gift from Google Research, and the Adobe Data Science Research award. Ganesh Ramakrishnan is grateful to IBM Research, India (specifically the IBM AI Horizon Networks - IIT Bombay initiative) as well as the IIT Bombay Institute Chair Professorship for their support and sponsorship. Abhishek Guttu and Aakriti are grateful to IBM AI Horizon Networks - IIT Bombay initiative. The views and conclusions contained herein are those of the authors and should not be interpreted as necessarily representing the official policies or endorsements, either expressed or implied, of the NSF, IBM Research, Google Research, Adobe Data Science or the U.S. government.

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
