# Supplementary Material

# Appendix

## Table of Contents

# A Code

The code of AUTOMATA is available at the following link: `https://anonymous.4open.science/r/AUTOMATA-F63C`.

# B Licenses

We release the code repository of AUTOMATA with MIT license, and it is available for everybody to use freely. We use the popular deep learning framework [41] for implementation of AUTOMATA framework, Ray-tune[36] for hyper-parameter search and scheduling algorithms, and CORDS [22] for subset selection strategies. As far as the datasets are considered, we use SST2 [47], SST5 [47], glue-SST2 [51], TREC6 [35, 18], CIFAR10 [28], SVHN [39], CIFAR100 [28], and DNA, SATIMAGE, LETTER, CONNECT-4 from **LIBSVM** (a library for Support Vector Machines (SVMs)) [7] datasets. CIFAR10, CIFAR100 datasets are released with an MIT license. SVHN dataset is released with a CC0:Public Domain license. Furthermore, all the datasets used in this work are publicly available. In addition, the datasets used do not contain any personally identifiable information.

# C AUTOMATA Algorithm Pseudocode

We give the pseudo code of AUTOMATA algorithm in Algorithm 1.

---
**Algorithm 1:** AUTOMATA Algorithm

---
**Input:** Hyper-parameter scheduler Algorithm: scheduler , Hyper-parameter search Algorithm: search , No. of configuration evaluations: $n$, Hyper-parameter search space: $\mathcal{H}$, Training dataset: $\mathcal{D}$, Validation dataset: $\mathcal{V}$, Total no of epochs: $T$, Epoch interval for subset selection: $R$, Size of the coreset: $k$, Reg. Coefficient: $\lambda$, Learning rates: $\{\alpha_t\}_{t=0}^{t=T-1}$, Tolerance: $\epsilon$

Generate $n$ configurations by calling the search algorithm
  $H = \{h_1, h_2, \cdots, h_n\} = \text{search}(\mathcal{H}, n)$
Randomly initialize each configuration model parameters $h_1.\theta = h_2.\theta = \cdots = h_n.\theta = \theta$
Set $h_1.t = h_2.t = \cdots = h_n.t = 0$;
Set $h_1.eval = h_2.eval = \cdots = h_n.eval = 0$;
Assign the initial resources(i.e., in our case training epochs) using the scheduler for all initialized
  configurations $\{h_i.r\}_{i=1}^{i=n} = \text{scheduler}(H, T)$
**repeat**
  ​  \*\*\*Evaluate all remaining configurations\*\*\*
  ​  **for** *each configuration numbered $i$ in $H$* **do**
  ​  ​  \*\*\*Train configuration $h_i$ using informative data subsets for $h_i.r$ epochs and evaluate on
  ​  ​  ​  validation set\*\*\*
  ​  ​  $h_i.eval, h_i.theta =$
  ​  ​  ​  subset-config-evaluation $(\mathcal{D}, \mathcal{V}, h_i.theta, h_i.r, R, k, \lambda, \{\alpha_t\}_{t=0}^{t=h_i.r}, \epsilon)$
  ​  ​  $h_i.t = h_i.t + h_i.r$
  ​  \*\*\*Assign resources again based on evaluation performance\*\*\*
  ​  $\{h_i.r\}_{i=1}^{i=n} = \text{scheduler}(H, T)$
**until** *until* $h_1.r == 0$ & $h_2.r == 0$ & $\cdots h_n.r == 0$
\*\*\*Get the best performing hyper-parameters based on final configuration evaluations\*\*\*
$finalconfig = \underset{h_i.config}{\text{argmax}}\, [h_i.eval]_{i=1}^n$
\*\*\*Perform final training using the best hyper-parameter configurations\*\*\*
$\theta_{final} = \text{finaltrain}(\theta, \mathcal{D}, finalconfig, T)$
**return** $\theta_{final}$

---

**Algorithm 2:** subset-config-evaluation

---

**Input:** Training dataset: $\mathcal{D}$, Validation dataset: $\mathcal{V}$, Initial model parameters: $\theta_0$, Total no of epochs: $T$, Epoch interval for subset selection: $R$, Size of the coreset: $k$, Reg. Coefficient: $\lambda$, Learning rates: $\{\alpha_t\}_{t=0}^{t=T-1}$, Tolerance: $\epsilon$

Set $t = 0$; Randomly initialize coreset $\mathcal{S}_0 \subseteq \mathcal{D} : |\mathcal{S}_0| = k$;

**repeat**

    **if** $(t\%R == 0) \wedge (t > 0)$ **then**

        $\mathcal{S}_t = \text{OMP}(\mathcal{D}, \theta_t, \lambda, \alpha_t, k, \epsilon)$

    **else**

        $\mathcal{S}_t = \mathcal{S}_{t-1}$

    Compute batches $\mathcal{D}_b = ((x_b, y_b); b \in (1 \cdots B))$ from $\mathcal{D}$

    Compute batches $\mathcal{S}_{tb} = ((x_b); b \in (1 \cdots B))$ from $\mathcal{S}$

    *** Mini-batch SGD ***

    Set $\theta_{t0} = \theta_t$

    **for** $b = 1$ *to* $B$ **do**

        Compute mask $\boldsymbol{m}_t$ on $\mathcal{S}_{tb}$ from current model parameters $\theta_{t(b-1)}$

        $\theta_{tb} = \theta_{t(b-1)} - \alpha_t \nabla_\theta L_S(\mathcal{D}_b, \theta_t) - \alpha_t \lambda_t \sum_{j \in \mathcal{S}_{tb}} \boldsymbol{m}_{jt} \nabla_\theta l_u(x_j, \theta_t(b-1))$

    Set $\theta_{t+1} = \theta_{tB}$

    $t = t + 1$

**until** *until* $t \geq T$

*** Evaluate trained model on validation set ***

$eval = \text{evaluate}\,(\theta_T, \mathcal{V})$

**return** $eval, \theta_T$

---

**Algorithm 3:** OMP

---

**Input:** Training loss $L_T$, current parameters: $\theta$, regularization coefficient: $\lambda$, subset size: $k$, tolerance: $\epsilon$

Initialize $\mathcal{S} = \emptyset$

$r \leftarrow \nabla_w(\| \sum_{l \in \mathcal{S}} \boldsymbol{w} \nabla_\theta L_T^l(\theta) - \nabla_\theta L_T(\theta)\| + \lambda \|\boldsymbol{w}\|^2 \|)_{\boldsymbol{w}=0}$

**repeat**

    $e = \text{argmax}_j |r_j|$

    $\mathcal{S} \leftarrow \mathcal{S} \cup \{e\}$

    $\boldsymbol{w} \leftarrow \text{argmin}_{\boldsymbol{w}}(\| \sum_{l \in \mathcal{S}} \boldsymbol{w} \nabla_\theta L_T^l(\theta) - \nabla_\theta L_T(\theta)\| + \lambda \|\boldsymbol{w}\|^2)$

    $r \leftarrow \nabla_{\boldsymbol{w}}(\| \sum_{l \in \mathcal{S}} \boldsymbol{w} \nabla_\theta L_T^l(\theta) - \nabla_\theta L_T(\theta)\| + \lambda \|\boldsymbol{w}\|^2 \|)$

**until** *until* $|\mathcal{S}| \leq k$ and $\| \sum_{l \in \mathcal{S}} \boldsymbol{w} \nabla_\theta L_T^l(\theta) - \nabla_\theta L_T(\theta)\| + \lambda \|\boldsymbol{w}\|^2 \geq \epsilon$

**return** $\mathcal{S}, \boldsymbol{w}$

---

## D  Per-Sample vs Per-Batch Subset Selection

We visualize the differences between Per-Sample and Per-Batch subset selection in Figure 5. In per-sample subset selection, we select a subset of data samples, whereas, in per-batch subset selection, we select a subset of mini-batches.

## E  More details on Hyper-parameter Search Algorithms

We give a brief overview of few representative hyper-parameter search algorithms, such as TPE [4] and Random Search [43] which we used in our experiments. As discussed earlier, given a hyper-parameter search space, hyper-parameter search algorithms provide a set of configurations that need to be evaluated. A naive way of performing the hyper-parameter search is Grid Search, which defines the search space as a grid and exhaustively evaluates each grid configuration. However, Grid Search is a time-consuming process, meaning that thousands to millions of configurations would need to be evaluated if the hyper-parameter space is large. In order to find optimal hyper-parameter settings quickly, Bayesian optimization-based hyper-parameter search algorithms have been developed. To

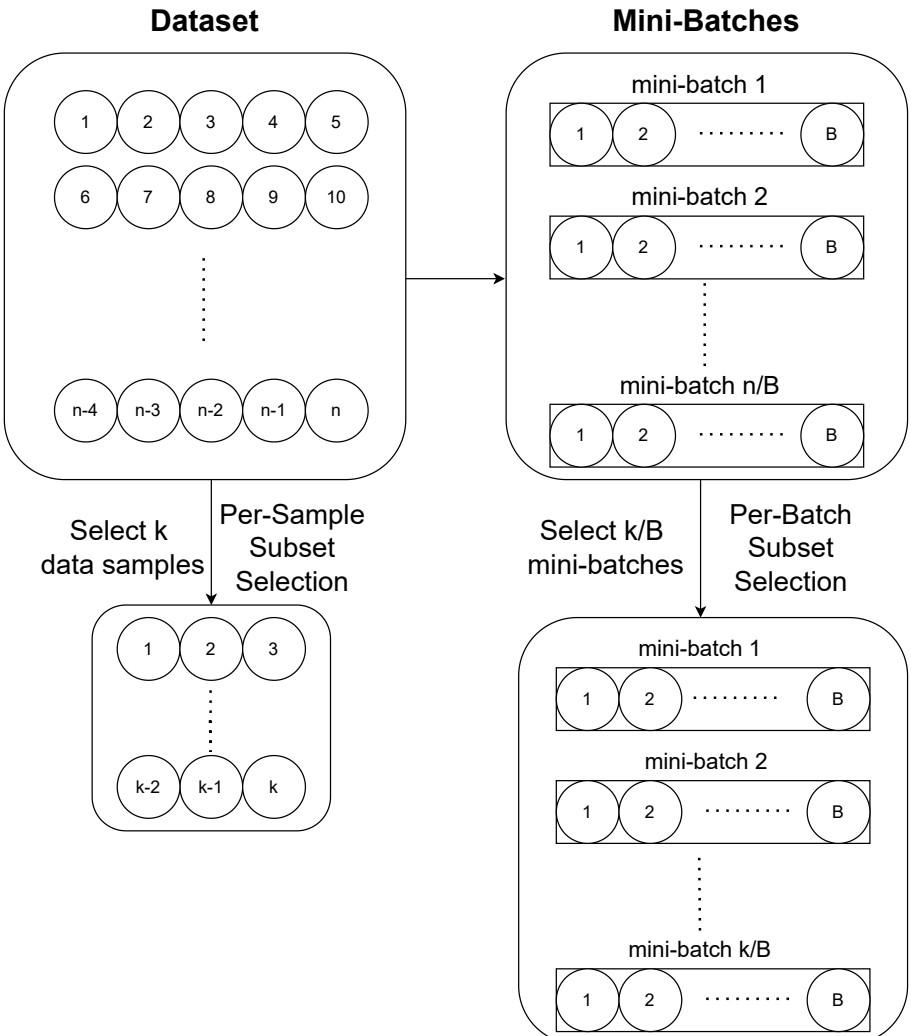

Figure 5: Visualization of Per-Sample vs Per-Batch Subset Selection

investigate the effectiveness of AUTOMATA across the spectrum of search algorithms, we used the Random Search method and the Bayesian optimization-based TPE method (described below) as representative hyper-parameter search algorithms.

### E.1   Random Search

In random search [43], hyper-parameter configurations are selected at random and evaluated to discover the optimal configuration among those chosen. As well as being more efficient than a grid search since it does not evaluate all possible configurations exhaustively, random search also reduces overfitting [2].

### E.2   Tree Parzen Structured Estimator (TPE)

TPE [4] is a sequential model-based optimization (SMBO) approach that sequentially constructs a probability model to approximate the performance of hyper-parameters based on historical configuration evaluations and then subsequently uses the model to select new configurations. TPE models the likelihood function $P(D|f)$ and the prior over the function space $P(f)$ using the kernel density estimation. TPE algorithm sorts the collected observations by the function evaluation value, typically validation set performance, and divides them into two groups based on some quantile. The first group $x_1$ contains best-performing observations, and the second group $x_2$ contains all other observations.

Then TPE models two different densities $i(x_1)$ and $g(x_2)$ based on the observations from the respective groups using kernel density estimation. Finally, TPE selects the subset observations that need to be evaluated by sampling from the distribution that models the maximum expected improvement, i.e., $\mathbb{E}[i(x)/g(x)]$.

## F   More details on Hyper-parameter Scheduling Algorithms

We give a brief overview of some representative hyper-parameter scheduling algorithms, such as HyperBand [32] and ASHA [34] which we used in our experiments. As discussed earlier, hyper-parameter scheduling algorithms improve the overall efficiency of the hyper-parameter tuning by terminating some of the poor configurations runs early. In our experiments, we consider Hyperband, and ASHA, which are extensions of the Sequential Halving algorithm (SHA) [20] that uses aggressive early stopping to terminate poor configuration runs and allocates an increasingly exponential amount of resources to the better performing configurations. SHA starts with $n$ number of initial configurations, each assigned with a minimum resource amount $r$. The SHA algorithm uses a reduction factor $\eta$ to reduce the number of configurations each round by selecting the top $\frac{1}{\eta}^{th}$ fraction of configurations while also increasing the resources allocated to these configurations by $\eta$ times each round. Following, we will discuss Hyperband and ASHA and the issues within SHA that each of them addresses.

### F.1   HyperBand

One of the issues with SHA is that its performance largely depends on the initial number $n$ of configurations. Hyperband [32] addresses this issue by performing a grid search over various feasible values of $n$. Further, each value of $n$ is associated with a minimum resource $r$ allocated to all configurations before some are terminated; larger values of $n$ are assigned smaller $r$ and hence more aggressive early-stopping. On the whole, in Hyperband [32] for different values of $n$ and $r$, the SHA algorithm is run until completion.

### F.2   ASHA:

One of the other issues with SHA is that the algorithm is sequential and has to wait for all the processes (assigned with an equal amount of resources) at a particular bracket to be completed before choosing the configurations to be selected for subsequent runs. Hence, due to the sequential nature of SHA, some GPU/CPU resources (with no processes running) cannot be effectively utilized in the distributed training setting, thereby taking more time for tuning. By contrast, ASHA [34] is an asynchronous variant of SHA and addresses the sequential issue of SHA by promoting a configuration to the next rung as long as there are GPU or CPU resources available. If no resources appear to be promotable, it randomly adds a new configuration to the base rung.

## G   More Experimental Details and Additional Results

### G.1   GPU Resources

We performed experiments on a mix of RTX 1080, RTX 2080, and V100 GPU servers containing 2-8 GPUs of 12GB memory. To be fair in timing computation, we ran AUTOMATA and all other baselines for a particular setting on the same GPU server.

### G.2   Additional Datasets Details

### G.2.1   Text Datasets

We performed experiments on SST2 [47], SST5 [47], glue-SST2 [51], and TREC6 [35, 18] text datasets. SST2 [47] and glue-SST2 [51] dataset classify the sentiment of the sentence (movie reviews) as negative or positive. Whereas SST5 [47] classify sentiment of sentence as negative, somewhat negative, neutral, somewhat positive or positive. TREC6 [35, 18] is a dataset for question classification consisting of open-domain, fact-based questions divided into broad semantic categories(ABBR - Abbreviation, DESC - Description and abstract concepts, ENTY - Entities, HUM - Human beings,

LOC - Locations, NYM - Numeric values). The train, text and validation splits for SST2 [47] and SST5 [47] are used from the source itself while the validation data for TREC6 [35, 18] is obtained using 10% of the train data. The test data for glue-SST2 [51] is obtained using 5% of the train data. Seed value of 42 is used in generator argument in random_split function of torch. In Table 1, we summarize the number classes, and number of instances in each split in the text datasets.

| Dataset | #Classes | #Train | #Validation | #Test |
|---------|----------|--------|-------------|-------|
| SST2 | 2 | 8544 | 1101 | 2210 |
| SST5 | 5 | 8544 | 1101 | 2210 |
| glue-SST2 | 2 | 63982 | 872 | 3367 |
| TREC6 | 6 | 4907 | 545 | 500 |

Table 1: Number of classes, Number of instances in Train, Validation and Test split in Text datasets

### G.2.2 Vision Datasets

We performed experiments on CIFAR10 [28], CIFAR100 [28], and SVHN [39] vision datasets. The CIFAR-10 [28] dataset contains 60,000 colored images of size $32 \times 32$ divided into ten classes, each with 6000 images. CIFAR100 [28] is also similar but that it has 600 images per class and 100 classes. Both CIFAR10 [28] and CIFAR100 [28] have 50,000 training samples and 10,000 test samples distributed equally across all classes. SVHN [39] is obtained from house numbers in Google Street View images and has 10 classes, one for each digit. The colored images of size $32 \times 32$ are centered around a single digit with some distracting characters on the side. SVHN [39] has 73,257 training digits, 26,032 testing digits. For all 3 datasets, $10\%$ of the training data is used for validation (seed value = 42). In Table 2, we summarize the number classes, and number of instances in each split in the image datasets.

| Dataset | #Classes | #Train | #Validation | #Test |
|---------|----------|--------|-------------|-------|
| CIFAR10 | 10 | 45000 | 5000 | 10000 |
| CIFAR100 | 100 | 45000 | 5000 | 10000 |
| SVHN | 10 | 65932 | 7325 | 26032 |

Table 2: Number of classes, Number of instances in Train, Validation and Test split in Image datasets

### G.2.3 Tabular Datasets

We performed experiments on the following tabular datasets **dna, letter, connect-4, and satimage** from **LIBSVM** (a library for Support Vector Machines (SVMs)) [7].

| Name | #Classes | #Train | #Validation | #Test | #Features |
|------|----------|--------|-------------|-------|-----------|
| dna | 3 | 1,400 | 600 | 1,186 | 180 |
| satimage | 6 | 3,104 | 1,331 | 2,000 | 36 |
| letter | 26 | 10,500 | 4,500 | 5,000 | 16 |
| connect_4 | 3 | 67,557 | - | - | 126 |

Table 3: Number of classes, Number of instances in Train, Validation and Test split in Tabular datasets

A brief description of the tabular datasets can be found in Table 3. For datasets without explicit validation and test datasets, 10% and 20% samples from the training set are used as validation and test datasets, respectively (seed value = 42).

## G.3 Ablation Studies

### G.3.1 Regularization Coefficient Ablation Study

We performed an ablation study to find the best regularization coefficient for data subset selection. In order to achieve this, we experimented on CIFAR10 dataset with the ResNet18 model and the same hyper-parameter search space used in the rest of the CIFAR10 experiments using only Random search

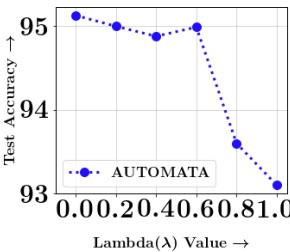

Figure 6: $\lambda$ Ablation Study

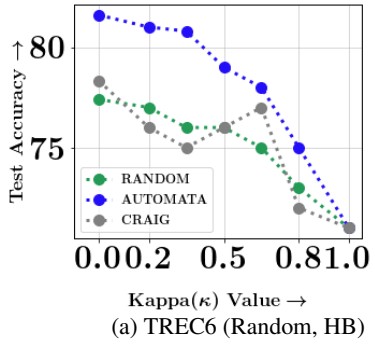

(a) TREC6 (Random, HB)

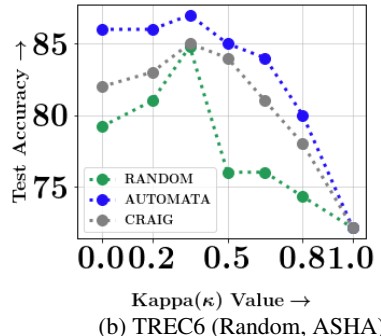

(b) TREC6 (Random, ASHA)

Figure 7: $\kappa$ Ablation Study

and no scheduler for lambda values of $0, 0.2, 0.4, 0.6, 0.8, 1$ respectively. We present the accuracies achieved by AUTOMATA at 10% subset for different values of lambda in Figure 6. From the results, it is evident that AUTOMATA achieved best performance when $\lambda = 0$. Hence, we used a $\lambda = 0$ in our experiments.

### G.3.2 Warm-Start Ablation Study

We performed an ablation study to find the best $\kappa$ value for warm-starting during model training. We experimented on TREC6 dataset with the LSTM model and the same hyper-parameter search space used in the rest of the TREC6 experiments for combinations of Random Search with ASHA and Hyperband schedulers for kappa values of $0, 0.2, 0.35, 0.5, 0.65, 0.8, 1$ respectively. We present the accuracies achieved by AUTOMATA at 10% subset for different values of kappa in Figure 7. Based on the results, we use $\kappa = 0.35$ with ASHA as scheduler and $\kappa = 0$ with Hyperband as scheduler.

We can explain the necessity of warm-starting with ASHA as a scheduler by the fact that the initial bracket occurs early (i.e., at $t = 1$) with ASHA. Accordingly, initial configuration evaluations made immediately after training for one epoch are used to promote configurations. There is a possibility that training such configurations on small data subsets may not be sufficient for making a sound decision about better-performing configurations. To prevent this, warm-starting for ASHA is necessary so that all configurations are trained on the entire data set for the first few epochs. With Hyperband, the brackets do not occur very early during training, so no warm-up is necessary.

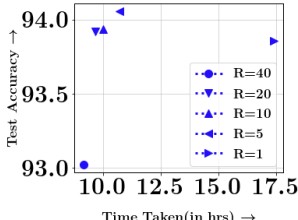

Figure 8: Performance of AUTOMATA with different $R$ values on CIFAR10 dataset using TPE+HyperBand

### G.3.3 Epoch Interval(R) Ablation Study

We performed an ablation study to find the best epoch interval value($R$) for data subset selection. To achieve this, we experimented on the CIFAR10 dataset with the ResNet18 model and the same hyper-parameter search space used in the rest of the CIFAR10 experiments using TPE and HyperBand as a scheduler for $R$ values of $1, 5, 10, 20, 40$ respectively. We present the accuracies achieved by AUTOMATA at 30% subset for different values of $R$ in Figure 8. From the results, it is evident that AUTOMATA achieved best performance vs efficiency tradeoff when $R = 20$. By using $R = 40$, a lower efficiency gain could be achieved at the expense of a greater performance loss. Using $R$ values of 1, 5, and 10, we achieve similar performance to $R = 20$ with a significant increase in tuning time. Therefore, our experiments used an epoch interval value of $R = 20$.

### G.4 Additional Experimental Details

For tuning with FULL datasets, the entire dataset is used to train the model during hyper-parameter tuning. But when the AUTOMATA (or CRAIG) is used, only a fraction of the dataset is used to train various models during tuning. Similar is the case with Random subset selection approach but the subsets are chosen at RANDOM. Note that subset selection techniques used are adaptive in nature, which mean that they chose subset every few epochs for the model to train on for coming few epochs.

### G.4.1 Details of Text Experiments

The hyper-parameter space for experiments on text datasets include learning rate, hidden size & number of layers of LSTM and batch size of training. Some experiments (with TPE search algorithm) where the best configuration among 27 configurations are found, the hyper-parameter space is learning rate: [0.001,0.1], LSTM hidden size: {64,128,256}, batch size: {16,32,64}. While the rest of the experiments where the best configuration among 54 configurations are found, the hyper-parameter space is learning rate: [0.001,0.1], LSTM hidden size: {64,128,256}, number of layers in LSTM: {1, 2}, batch size: {16,32,64}.

### G.4.2 Details of Image Experiments

The hyper-parameter search space for tuning experiments on image datasets include a choice between Momentum method and Nesterov Accelerated Gradient method, choice of learning rate scheduler and their corresponding parameters, and four different group-wise learning rates, $lr_1$ for layers of the first group, $lr_2$ for layers of intermediate groups, $lr_3$ for layers of the last group of ResNet model, and $lr_4$ for the final fully connected layer. For learning rate scheduler, we change the learning rates during training using either a cosine annealing schedule or decay it linearly by $\gamma$ after every 20 epochs. Best configuration for most experiments is selected from 27 configurations where the hyper-parameter space is $lr_1$: [0.001, 0.01], $lr_2$: [0.001, 0.01], $lr_3$: [0.001, 0.01], $lr_4$: [0.001, 0.01], Nesterov: {True, False}, learning rate scheduler: {Cosine Annealing, Linear Decay}, $\gamma$: [0.05, 0.5].

### G.4.3 Details of Tabular Experiments

The hyper-parameter search space consists of a choice between the Stochastic Gradient Descent(SGD) optimizer or Adam optimizer, choice of learning rate $lr$, choice of learning rate scheduler, the sizes of the two hidden layers $h_1$ and $h_2$ and batch size for training. For learning rate scheduler, we either don't use a learning rate scheduler or change the learning rates during training using a cosine annealing schedule or decay it linearly by 0.05 after every 20 epochs. Best configuration for most experiments is selected from 27 configurations where the hyper-parameter space is $lr$: [0.001, 0.01], Optimizer: {Adam, SGD}, learning rate scheduler: {None, Cosine Annealing, Linear Decay}, $h_1$: {150, 200, 250, 300}, $h_2$: {150, 200, 250, 300} and batch size: {16,32,64}.

### G.5 More Hyper-parameter Tuning Results

We present more hyper-parameter tuning results of AUTOMATA on additional text, image, and tabular datasets in Figures 9,10,11. We also present Average Wall Clock times taken by AUTOMATA for tuning on text, image, and tabular datasets in Tables 7,8,9. From the results, it is evident that AUTOMATA achieves best speedup vs. accuracy tradeoff in almost all of the cases.

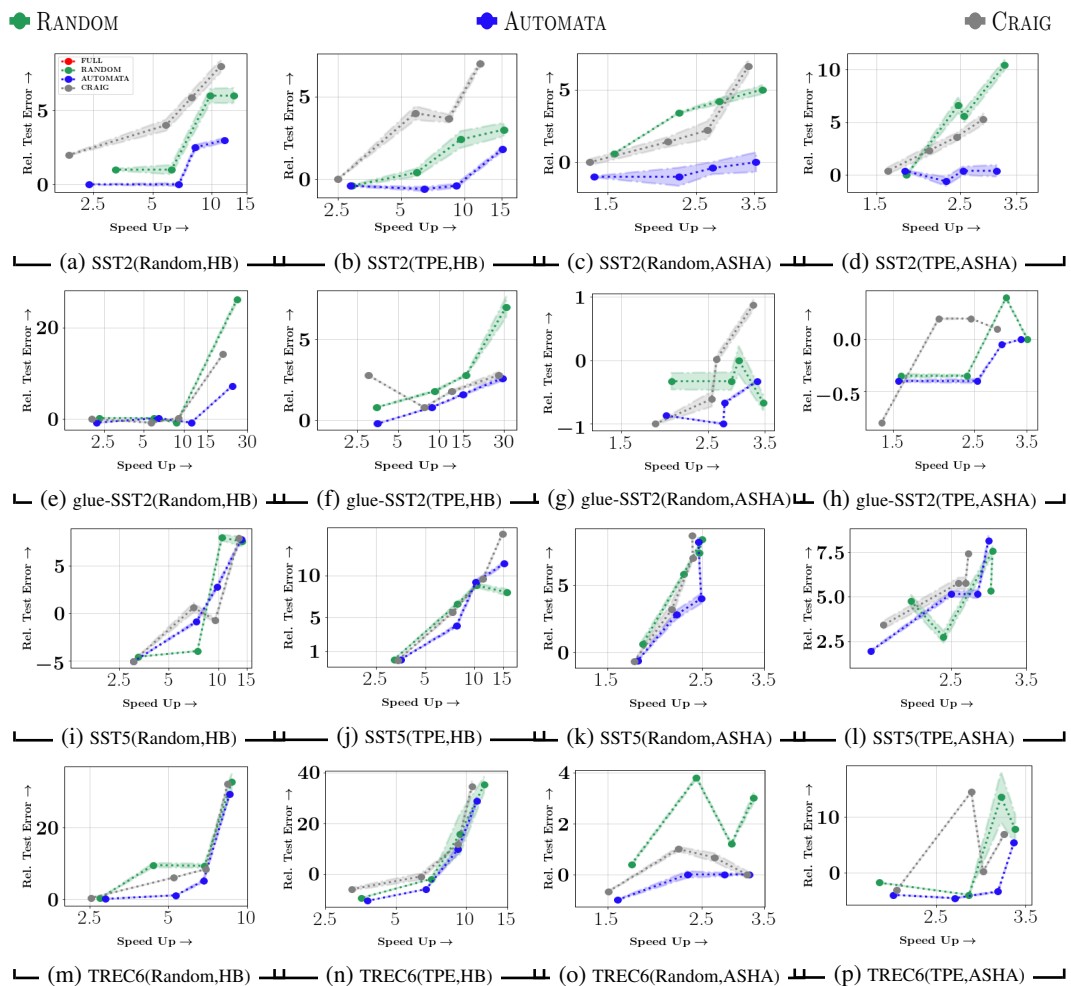

Figure 9: Tuning Results on Text Datasets: Comparison of performance of AUTOMATA with baselines(RANDOM, CRAIG, FULL) for Hyper-parameter tuning. In sub-figures (a-p), we present speedup *vs.* relative test error (in %), compared to Full data tuning for different methods. On each scatter plot, smaller subsets appear on the right, and larger ones appear on the left. Results are shown for (a-d) SST2, (e-h) glue-SST2, (i-l) SST5, (m-p) TREC6 datasets with different combinations of hyper-parameter search and scheduling algorithms. *The scatter plots show that* AUTOMATA *achieves the best speedup-accuracy tradeoff in almost every case (**bottom-right corner of each plot indicates the best speedup-accuracy tradeoff region**).*

### G.5.1 Standard Deviations

We present the standard deviations for text experiments in Table 4, for image experiments in Table 5, and for tabular experiments in Table 6.

### G.5.2 Average Wall Clock Timings

We present the average wall clock timing in seconds for text experiments in Table 7, for image experiments in Table 8, and for tabular experiments in Table 9.

### G.6 CO2 Emissions and Energy Consumption Results

Sub-figures 12a,12b,12c,12d shows the energy efficiency plot of AUTOMATA on CIFAR100 dataset for 1%, 5%, 10%, 30% subset fractions. For calculating the energy consumed by the GPU/CPU cores, we use pyJoules[6]. From the plot, it is evident that AUTOMATA is more energy efficient compared to

---

[6] https://pypi.org/project/pyJoules/.

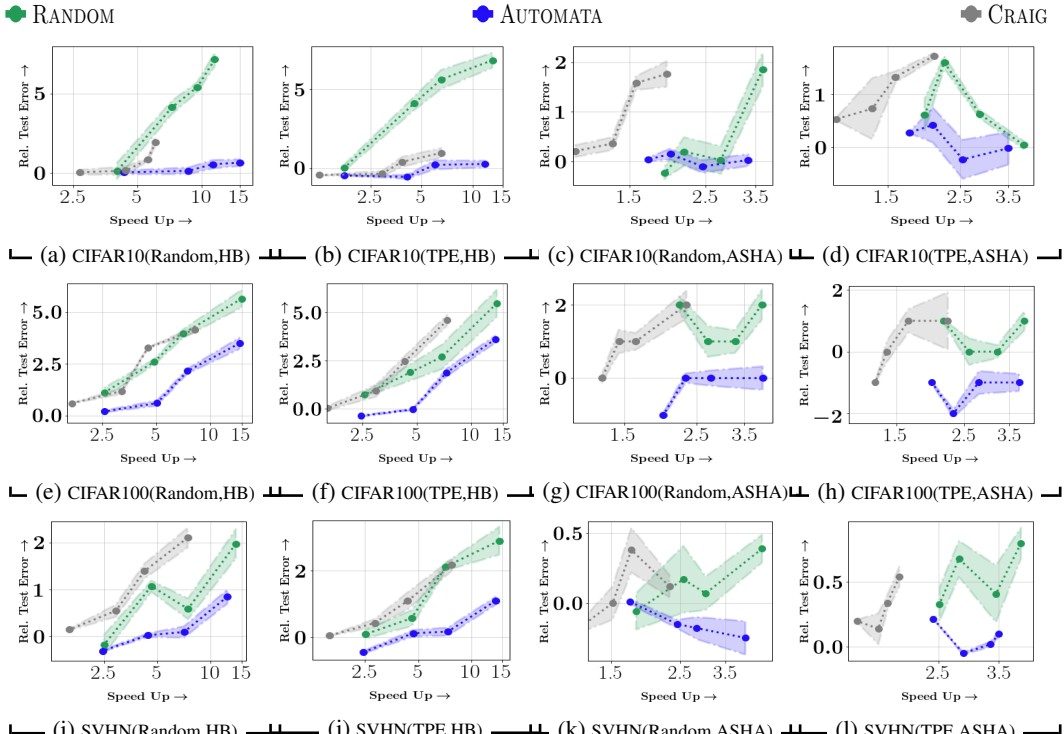

Figure 10: Tuning Results on Image Datasets: Comparison of performance of AUTOMATA with baselines(RANDOM, CRAIG, FULL) for Hyper-parameter tuning. In sub-figures (a-l), we present speedup *vs.* relative test error (in %), compared to Full data tuning for different methods. On each scatter plot, smaller subsets appear on the right, and larger ones appear on the left. Results are shown for (a-d) CIFAR10, (e-h) CIFAR100, (i-l) SVHN datasets with different combinations of hyper-parameter search and scheduling algorithms. *The scatter plots show that* AUTOMATA *achieves the best speedup-accuracy tradeoff in almost every case (**bottom-right corner of each plot indicates the best speedup-accuracy tradeoff region**).*

the other baselines and full data tuning. Sub-figures 12e,12f,12g,12h shows the plot of relative error vs $CO_2$ emissions efficiency, both w.r.t full training. $CO_2$ emissions were estimated based on the total compute time using the https://mlco2.github.io/impact#computeMachine Learning Impact calculator presented in [30]. From the results, it is evident that AUTOMATA achieved the best energy vs. accuracy tradeoff and is environmentally friendly based on $CO_2$ emissions compared to other baselines (including CRAIG and RANDOM).

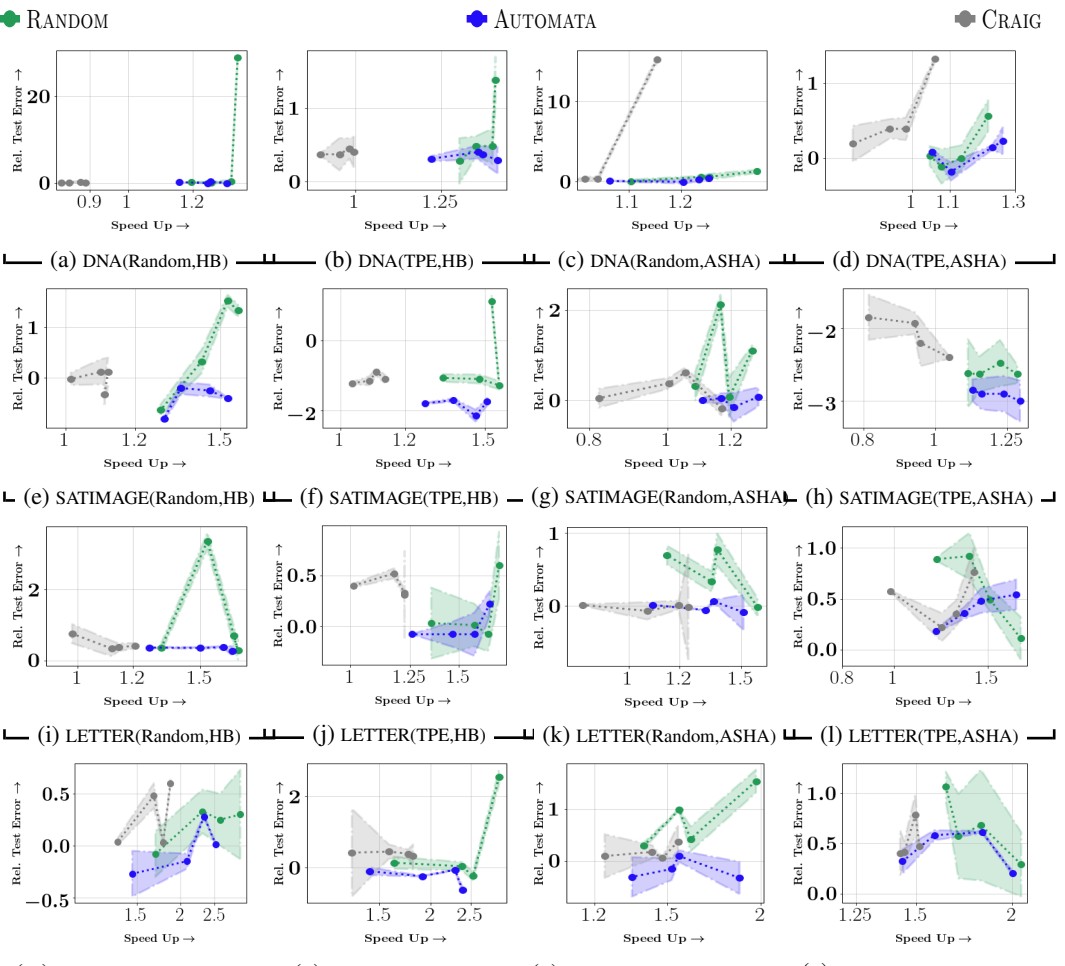

Figure 11: Tuning Results on Tabular Datasets: Comparison of performance of AUTOMATA with baselines(RANDOM, CRAIG, FULL) for Hyper-parameter tuning. In sub-figures (a-p), we present speedup *vs.* relative test error (in %), compared to Full data tuning for different methods. On each scatter plot, smaller subsets appear on the right, and larger ones appear on the left. Results are shown for (a-d) DNA, (e-h) SATIMAGE, (i-l) LETTER, (m-p) CONNECT-4 datasets with different combinations of hyper-parameter search and scheduling algorithms. *The scatter plots show that* AUTOMATA *achieves the best speedup-accuracy tradeoff in almost every case (**bottom-right corner of each plot indicates the best speedup-accuracy tradeoff region**).*

| Standard Deviation Results | | | | | | |
|---|---|---|---|---|---|---|
| Search Algorithm, Scheduler | | | TPE, HyperBand | | | |
| | | | Standard deviation of the Model(for 5 runs) | | | |
| | | Budget(%) | 1% | 5% | 10% | 30% |
| Dataset | Model | Selection Strategy | | | | |
| SST2 | LSTM | FULL | 0.003 | 0.003 | 0.003 | 0.003 |
| | | RANDOM | 0.41 | 0.51 | 0.24 | 0.05 |
| | | CRAIG | 0.21 | 0.24 | 0.42 | 0.02 |
| | | AUTOMATA | 0.13 | 0.12 | 0.13 | 0.04 |
| SST5 | LSTM | FULL | 0.02 | 0.02 | 0.02 | 0.02 |
| | | RANDOM | 0.38 | 0.24 | 0.18 | 0.01 |
| | | CRAIG | 0.01 | 0.14 | 0.19 | 0.06 |
| | | AUTOMATA | 0.17 | 0.27 | 0.19 | 0.02 |
| glue-SST2 | LSTM | FULL | 0.01 | 0.01 | 0.01 | 0.01 |
| | | RANDOM | 0.63 | 0.07 | 0.06 | 0.03 |
| | | CRAIG | 0.21 | 0.06 | 0.0 | 0.0 |
| | | AUTOMATA | 0.15 | 0.05 | 0.03 | 0.0 |
| TREC6 | LSTM | FULL | 0.04 | 0.04 | 0.04 | 0.04 |
| | | RANDOM | 3.13 | 7.99 | 0.07 | 0.07 |
| | | CRAIG | 1.94 | 0.15 | 1.29 | 0.62 |
| | | AUTOMATA | 1.49 | 1.77 | 0.02 | 0.02 |
| Search Algorithm, Scheduler | | | Random, HyperBand | | | |
| | | | Standard deviation of the Model(for 5 runs) | | | |
| | | Budget(%) | 1% | 5% | 10% | 30% |
| Dataset | Model | Selection Strategy | | | | |
| SST2 | LSTM | FULL | 0.002 | 0.002 | 0.002 | 0.002 |
| | | RANDOM | 0.52 | 0.48 | 0.32 | 0.07 |
| | | CRAIG | 0.43 | 0.41 | 0.38 | 0.04 |
| | | AUTOMATA | 0.14 | 0.18 | 0.10 | 0.03 |
| SST5 | LSTM | FULL | 0.19 | 0.19 | 0.19 | 0.19 |
| | | RANDOM | 0.52 | 0.31 | 0.03 | 0.01 |
| | | CRAIG | 0.34 | 0.17 | 0.5 | 0.02 |
| | | AUTOMATA | 0.21 | 0.21 | 0.15 | 0.02 |
| glue-SST2 | LSTM | FULL | 0.01 | 0.01 | 0.01 | 0.01 |
| | | RANDOM | 0.54 | 0.21 | 0.04 | 0.03 |
| | | CRAIG | 0.18 | 0.13 | 0.08 | 0.02 |
| | | AUTOMATA | 0.12 | 0.04 | 0.0 | 0.0 |
| TREC6 | LSTM | FULL | 0.01 | 0.01 | 0.01 | 0.01 |
| | | RANDOM | 2.5 | 0.6 | 0.6 | 0.04 |
| | | CRAIG | 0.54 | 0.08 | 0.0 | 0.01 |
| | | AUTOMATA | 0.49 | 0.45 | 0.06 | 0.04 |
| Search Algorithm, Scheduler | | | TPE, ASHA | | | |
| | | | Standard deviation of the Model(for 5 runs) | | | |
| | | Budget(%) | 1% | 5% | 10% | 30% |
| Dataset | Model | Selection Strategy | | | | |
| SST2 | LSTM | FULL | 0.55 | 0.55 | 0.55 | 0.55 |
| | | RANDOM | 0.51 | 0.57 | 0.73 | 0.03 |
| | | CRAIG | 0.52 | 0.38 | 0.48 | 0.32 |
| | | AUTOMATA | 0.55 | 0.4 | 0.43 | 0.05 |
| SST5 | LSTM | FULL | 0.02 | 0.02 | 0.02 | 0.02 |
| | | RANDOM | 0.26 | 0.28 | 0.33 | 0.35 |
| | | CRAIG | 0.0 | 0.33 | 0.32 | 0.22 |
| | | AUTOMATA | 0.26 | 0.23 | 0.15 | 0.05 |
| glue-SST2 | LSTM | FULL | 0.01 | 0.01 | 0.01 | 0.01 |
| | | RANDOM | 0.01 | 0.01 | 0.01 | 0.02 |
| | | CRAIG | 0.00 | 0.000 | 0.01 | 0.01 |
| | | AUTOMATA | 0.00 | 0.00 | 0.02 | 0.01 |
| TREC6 | LSTM | FULL | 0.39 | 0.39 | 0.39 | 0.39 |
| | | RANDOM | 2.68 | 4.55 | 0.01 | 0.04 |
| | | CRAIG | 0.23 | 0.01 | 0.01 | 0.02 |
| | | AUTOMATA | 0.0 | 0.14 | 0.08 | 0.01 |
| Search Algorithm, Scheduler | | | Random, ASHA | | | |
| | | | Standard deviation of the Model(for 5 runs) | | | |
| | | Budget(%) | 1% | 5% | 10% | 30% |
| Dataset | Model | Selection Strategy | | | | |
| SST2 | LSTM | FULL | 0.13 | 0.13 | 0.13 | 0.13 |
| | | RANDOM | 0.53 | 0.21 | 0.04 | 0.02 |
| | | CRAIG | 0.46 | 0.62 | 0.34 | 0.21 |
| | | AUTOMATA | 0.68 | 0.42 | 0.63 | 0.05 |
| SST5 | LSTM | FULL | 0.12 | 0.12 | 0.12 | 0.12 |
| | | RANDOM | 0.08 | 0.08 | 0.12 | 0.31 |
| | | CRAIG | 0.06 | 0.11 | 0.58 | 0.02 |
| | | AUTOMATA | 0.31 | 0.37 | 0.33 | 0.02 |
| glue-SST2 | LSTM | FULL | 0.01 | 0.01 | 0.01 | 0.01 |
| | | RANDOM | 0.12 | 0.22 | 0.14 | 0.13 |
| | | CRAIG | 0.08 | 0.06 | 0.12 | 0.01 |
| | | AUTOMATA | 0.02 | 0.02 | 0.01 | 0.00 |
| TREC6 | LSTM | FULL | 0.15 | 0.15 | 0.15 | 0.15 |
| | | RANDOM | 0.2 | 0.04 | 0.07 | 0.06 |
| | | CRAIG | 0.14 | 0.17 | 0.09 | 0.02 |
| | | AUTOMATA | 0.01 | 0.07 | 0.13 | 0.03 |

Table 4: Standard deviation results for SST2, SST5, glue-SST2 and TREC6 datasets for 5 runs

| Standard Deviation Results | | | | | | |
|---|---|---|---|---|---|---|
| Search Algorithm, Scheduler | | | TPE, HyperBand | | | |
| | | | Standard deviation of the Model(for 3 runs) | | | |
| | | Budget(%) | 1% | 5% | 10% | 30% |
| Dataset | Model | Selection Strategy | | | | |
| CIFAR100 | ResNet18 | FULL | 0.057 | 0.057 | 0.057 | 0.057 |
| | | RANDOM | 0.71 | 0.64 | 0.43 | 0.12 |
| | | CRAIG | 0.28 | 0.29 | 0.31 | 0.11 |
| | | AUTOMATA | 0.19 | 0.21 | 0.04 | 0.04 |
| CIFAR10 | ResNet18 | FULL | 0.032 | 0.032 | 0.032 | 0.032 |
| | | RANDOM | 0.48 | 0.61 | 0.43 | 0.21 |
| | | CRAIG | 0.31 | 0.32 | 0.18 | 0.02 |
| | | AUTOMATA | 0.21 | 0.29 | 0.13 | 0.02 |
| SVHN | ResNet18 | FULL | 0.012 | 0.012 | 0.012 | 0.012 |
| | | RANDOM | 0.42 | 0.12 | 0.24 | 0.13 |
| | | CRAIG | 0.18 | 0.26 | 0.12 | 0.01 |
| | | AUTOMATA | 0.12 | 0.12 | 0.10 | 0.04 |
| Search Algorithm, Scheduler | | | Random, HyperBand | | | |
| | | | Standard deviation of the Model(for 3 runs) | | | |
| | | Budget(%) | 1% | 5% | 10% | 30% |
| Dataset | Model | Selection Strategy | | | | |
| CIFAR100 | ResNet18 | FULL | 0.054 | 0.054 | 0.054 | 0.054 |
| | | RANDOM | 0.43 | 0.21 | 0.14 | 0.24 |
| | | CRAIG | 0.16 | 0.02 | 0.14 | 0.02 |
| | | AUTOMATA | 0.28 | 0.12 | 0.13 | 0.05 |
| CIFAR10 | ResNet18 | FULL | 0.039 | 0.039 | 0.039 | 0.039 |
| | | RANDOM | 0.43 | 0.32 | 0.42 | 0.51 |
| | | CRAIG | 0.26 | 0.11 | 0.258 | 0.16 |
| | | AUTOMATA | 0.21 | 0.27 | 0.213 | 0.12 |
| SVHN | ResNet18 | FULL | 0.021 | 0.021 | 0.021 | 0.021 |
| | | RANDOM | 0.31 | 0.22 | 0.12 | 0.21 |
| | | CRAIG | 0.21 | 0.16 | 0.13 | 0.02 |
| | | AUTOMATA | 0.15 | 0.12 | 0.04 | 0.02 |
| Search Algorithm, Scheduler | | | TPE, ASHA | | | |
| | | | Standard deviation of the Model(for 3 runs) | | | |
| | | Budget(%) | 1% | 5% | 10% | 30% |
| Dataset | Model | Selection Strategy | | | | |
| CIFAR100 | ResNet18 | FULL | 0.12 | 0.12 | 0.12 | 0.12 |
| | | RANDOM | 0.31 | 0.21 | 0.43 | 0.038 |
| | | CRAIG | 0.93 | 0.38 | 0.34 | 0.021 |
| | | AUTOMATA | 0.28 | 0.37 | 0.13 | 0.05 |
| CIFAR10 | ResNet18 | FULL | 0.12 | 0.12 | 0.12 | 0.12 |
| | | RANDOM | 0.08 | 0.08 | 0.12 | 0.31 |
| | | CRAIG | 0.06 | 0.11 | 0.58 | 0.02 |
| | | AUTOMATA | 0.31 | 0.37 | 0.33 | 0.02 |
| SVHN | ResNet18 | FULL | 0.01 | 0.01 | 0.01 | 0.01 |
| | | RANDOM | 0.12 | 0.22 | 0.14 | 0.13 |
| | | CRAIG | 0.08 | 0.06 | 0.12 | 0.01 |
| | | AUTOMATA | 0.02 | 0.02 | 0.01 | 0.00 |
| Search Algorithm, Scheduler | | | Random, ASHA | | | |
| | | | Standard deviation of the Model(for 3 runs) | | | |
| | | Budget(%) | 1% | 5% | 10% | 30% |
| Dataset | Model | Selection Strategy | | | | |
| CIFAR100 | ResNet18 | FULL | 0.14 | 0.14 | 0.14 | 0.14 |
| | | RANDOM | 0.41 | 0.31 | 0.42 | 0.23 |
| | | CRAIG | 0.38 | 0.23 | 0.32 | 0.12 |
| | | AUTOMATA | 0.32 | 0.21 | 0.13 | 0.05 |
| CIFAR10 | ResNet18 | FULL | 0.08 | 0.08 | 0.08 | 0.08 |
| | | RANDOM | 0.34 | 0.28 | 0.29 | 0.12 |
| | | CRAIG | 0.26 | 0.14 | 0.13 | 0.12 |
| | | AUTOMATA | 0.12 | 0.13 | 0.11 | 0.04 |
| SVHN | ResNet18 | FULL | 0.04 | 0.04 | 0.04 | 0.04 |
| | | RANDOM | 0.10 | 0.12 | 0.24 | 0.13 |
| | | CRAIG | 0.08 | 0.16 | 0.12 | 0.03 |
| | | AUTOMATA | 0.12 | 0.08 | 0.03 | 0.02 |

Table 5: Standard deviation results for CIFAR100, CIFAR10, and SVHN datasets for 3 runs

| Standard Deviation Results | | | | | | |
|---|---|---|---|---|---|---|
| Search Algorithm, Scheduler | | | TPE, HyperBand | | | |
| | | | Standard deviation of the Model(for 5 runs) | | | |
| | | Budget(%) | 1% | 5% | 10% | 30% |
| Dataset | Model | Selection Strategy | | | | |
| DNA | MLP | FULL | 0.05 | 0.05 | 0.05 | 0.05 |
| | | RANDOM | 0.32 | 0.24 | 0.14 | 0.31 |
| | | CRAIG | 0.21 | 0.14 | 0.22 | 0.02 |
| | | AUTOMATA | 0.18 | 0.12 | 0.09 | 0.04 |
| SATIMAGE | MLP | FULL | 0.02 | 0.02 | 0.02 | 0.02 |
| | | RANDOM | 0.18 | 0.12 | 0.13 | 0.11 |
| | | CRAIG | 0.12 | 0.12 | 0.13 | 0.04 |
| | | AUTOMATA | 0.11 | 0.17 | 0.05 | 0.02 |
| LETTER | MLP | FULL | 0.065 | 0.065 | 0.065 | 0.065 |
| | | RANDOM | 0.33 | 0.17 | 0.21 | 0.35 |
| | | CRAIG | 0.431 | 0.036 | 0.05 | 0.02 |
| | | AUTOMATA | 0.124 | 0.213 | 0.123 | 0.003 |
| CONNECT-4 | MLP | FULL | 0.021 | 0.021 | 0.021 | 0.021 |
| | | RANDOM | 0.21 | 0.12 | 0.12 | 0.17 |
| | | CRAIG | 0.33 | 0.121 | 0.20 | 1.23 |
| | | AUTOMATA | 0.09 | 0.017 | 0.012 | 0.11 |
| Search Algorithm, Scheduler | | | Random, HyperBand | | | |
| | | | Standard deviation of the Model(for 5 runs) | | | |
| | | Budget(%) | 1% | 5% | 10% | 30% |
| Dataset | Model | Selection Strategy | | | | |
| DNA | MLP | FULL | 0.042 | 0.042 | 0.042 | 0.042 |
| | | RANDOM | 0.212 | 0.431 | 0.214 | 0.124 |
| | | CRAIG | 0.127 | 0.201 | 0.210 | 0.12 |
| | | AUTOMATA | 0.091 | 0.012 | 0.101 | 0.014 |
| SATIMAGE | MLP | FULL | 0.021 | 0.021 | 0.021 | 0.021 |
| | | RANDOM | 0.12 | 0.12 | 0.23 | 0.11 |
| | | CRAIG | 0.21 | 0.28 | 0.25 | 0.122 |
| | | AUTOMATA | 0.01 | 0.12 | 0.12 | 0.062 |
| LETTER | MLP | FULL | 0.054 | 0.054 | 0.054 | 0.054 |
| | | RANDOM | 0.18 | 0.26 | 0.234 | 0.13 |
| | | CRAIG | 0.08 | 0.11 | 0.22 | 0.27 |
| | | AUTOMATA | 0.13 | 0.034 | 0.021 | 0.023 |
| CONNECT-4 | MLP | FULL | 0.031 | 0.031 | 0.031 | 0.031 |
| | | RANDOM | 0.43 | 0.25 | 0.21 | 0.22 |
| | | CRAIG | 0.037 | 0.028 | 0.12 | 0.034 |
| | | AUTOMATA | 0.0148 | 0.015 | 0.074 | 0.22 |
| Search Algorithm, Scheduler | | | TPE, ASHA | | | |
| | | | Standard deviation of the Model(for 5 runs) | | | |
| | | Budget(%) | 1% | 5% | 10% | 30% |
| Dataset | Model | Selection Strategy | | | | |
| DNA | MLP | FULL | 0.47 | 0.47 | 0.47 | 0.47 |
| | | RANDOM | 0.21 | 0.17 | 0.23 | 0.13 |
| | | CRAIG | 0.03 | 0.15 | 0.13 | 0.24 |
| | | AUTOMATA | 0.18 | 0.03 | 0.12 | 0.035 |
| SATIMAGE | MLP | FULL | 0.132 | 0.132 | 0.132 | 0.132 |
| | | RANDOM | 0.126 | 0.318 | 0.243 | 0.47 |
| | | CRAIG | 0.02 | 0.321 | 0.152 | 0.31 |
| | | AUTOMATA | 0.282 | 0.245 | 0.215 | 0.15 |
| LETTER | MLP | FULL | 0.11 | 0.11 | 0.11 | 0.11 |
| | | RANDOM | 0.21 | 0.12 | 0.23 | 0.02 |
| | | CRAIG | 0.24 | 0.13 | 0.14 | 0.01 |
| | | AUTOMATA | 0.15 | 0.12 | 0.06 | 0.02 |
| CONNECT-4 | MLP | FULL | 0.25 | 0.25 | 0.25 | 0.25 |
| | | RANDOM | 0.32 | 0.55 | 0.42 | 0.15 |
| | | CRAIG | 0.021 | 0.21 | 0.11 | 0.23 |
| | | AUTOMATA | 0.02 | 0.013 | 0.05 | 0.12 |
| Search Algorithm, Scheduler | | | Random, ASHA | | | |
| | | | Standard deviation of the Model(for 5 runs) | | | |
| | | Budget(%) | 1% | 5% | 10% | 30% |
| Dataset | Model | Selection Strategy | | | | |
| DNA | MLP | FULL | 0.13 | 0.13 | 0.13 | 0.13 |
| | | RANDOM | 0.21 | 0.37 | 0.13 | 0.21 |
| | | CRAIG | 0.38 | 0.31 | 0.21 | 0.011 |
| | | AUTOMATA | 0.34 | 0.21 | 0.23 | 0.05 |
| SATIMAGE | MLP | FULL | 0.24 | 0.24 | 0.24 | 0.24 |
| | | RANDOM | 0.12 | 0.438 | 0.22 | 0.32 |
| | | CRAIG | 0.16 | 0.11 | 0.12 | 0.21 |
| | | AUTOMATA | 0.21 | 0.32 | 0.11 | 0.054 |
| LETTER | MLP | FULL | 0.032 | 0.032 | 0.032 | 0.032 |
| | | RANDOM | 0.12 | 0.22 | 0.14 | 0.13 |
| | | CRAIG | 0.72 | 0.06 | 0.12 | 0.01 |
| | | AUTOMATA | 0.24 | 0.02 | 0.01 | 0.00 |
| CONNECT-4 | MLP | FULL | 0.152 | 0.152 | 0.152 | 0.152 |
| | | RANDOM | 0.221 | 0.24 | 0.021 | 0.06 |
| | | CRAIG | 0.321 | 0.015 | 0.132 | 0.42 |
| | | AUTOMATA | 0.311 | 0.112 | 0.214 | 0.38 |

Table 6: Standard deviation results for DNA, SATIMAGE, LETTER, and CONNECT-4 datasets for 5 runs

| Average Wall Clock Time Results | | | | | | |
|---|---|---|---|---|---|---|
| **Search Algorithm, Scheduler** | | | **TPE, HyperBand** | | | |
| | | | Average Wall Clock Time in seconds(for 5 runs) | | | |
| | | Budget(%) | 1% | 5% | 10% | 30% |
| Dataset | Model | Selection Strategy | | | | |
| SST2 | LSTM | FULL | 22560.43 | 22560.43 | 22560.43 | 22560.43 |
| | | RANDOM | 1462.84 | 2345.51 | 3807.84 | 7693.05 |
| | | CRAIG | 1490.31 | 2454.13 | 3503.55 | 7831.03 |
| | | AUTOMATA | 1900.04 | 2669.64 | 3860.69 | 8961.92 |
| SST5 | LSTM | FULL | 225211.4 | 225211.4 | 225211.4 | 225211.4 |
| | | RANDOM | 7207.79 | 14215.17 | 24308.84 | 65149.43 |
| | | CRAIG | 7670.41 | 15090.77 | 25525.25 | 63867.59 |
| | | AUTOMATA | 8166.06 | 18130.39 | 29022.47 | 75243.94 |
| glue-SST2 | LSTM | FULL | 27363.23 | 27363.23 | 27363.23 | 27363.23 |
| | | RANDOM | 1740.62 | 2654 | 3492.29 | 8473.01 |
| | | CRAIG | 1798.09 | 2685.41 | 3524.4 | 7740.29 |
| | | AUTOMATA | 1851.64 | 2430.48 | 3729.37 | 8033.01 |
| TREC6 | LSTM | FULL | 18305.66 | 18305.66 | 18305.66 | 18305.66 |
| | | RANDOM | 1531.33 | 1943.72 | 2583.14 | 5142.32 |
| | | CRAIG | 1646.82 | 1979.17 | 2711.13 | 4826.62 |
| | | AUTOMATA | 1726.13 | 1987.16 | 2853.45 | 5626.3 |
| **Search Algorithm, Scheduler** | | | **Random, HyperBand** | | | |
| | | | Average Wall Clock Time in seconds(for 5 runs) | | | |
| | | Budget(%) | 1% | 5% | 10% | 30% |
| Dataset | Model | Selection Strategy | | | | |
| SST2 | LSTM | FULL | 22518.35 | 22518.35 | 22518.35 | 22518.35 |
| | | RANDOM | 1746.98 | 2294.75 | 3613.07 | 6947.87 |
| | | CRAIG | 1941.63 | 2738.63 | 3313.03 | 9499.05 |
| | | AUTOMATA | 2031.04 | 2849.43 | 3860.69 | 11961.92 |
| SST5 | LSTM | FULL | 226102.3 | 226102.3 | 226102.3 | 226102.3 |
| | | RANDOM | 8959.24 | 25550.53 | 38103.64 | 97729.49 |
| | | CRAIG | 9787.63 | 19847.2 | 35104.98 | 102625.02 |
| | | AUTOMATA | 11432.15 | 24742.1 | 39615.83 | 111391.12 |
| glue-SST2 | LSTM | FULL | 49301.14 | 49301.14 | 49301.14 | 49301.14 |
| | | RANDOM | 3539.39 | 4691.9 | 6602.08 | 15097.96 |
| | | CRAIG | 3571.58 | 5011.42 | 6702.92 | 16134.73 |
| | | AUTOMATA | 3689.26 | 5139.68 | 6951.21 | 16253.83 |
| TREC6 | LSTM | FULL | 26443.49 | 26443.49 | 26443.49 | 26443.49 |
| | | RANDOM | 3027.98 | 3846.24 | 6048.7 | 9653.66 |
| | | CRAIG | 3083.63 | 3885.47 | 4948.91 | 9208.35 |
| | | AUTOMATA | 3139.64 | 3805.95 | 5052.73 | 10462.96 |
| **Search Algorithm, Scheduler** | | | **TPE, ASHA** | | | |
| | | | Average Wall Clock Time in seconds(for 5 runs) | | | |
| | | Budget(%) | 1% | 5% | 10% | 30% |
| Dataset | Model | Selection Strategy | | | | |
| SST2 | LSTM | FULL | 7979.37 | 7979.37 | 7979.37 | 7979.37 |
| | | RANDOM | 2447.21 | 3122.28 | 3239.47 | 4424.17 |
| | | CRAIG | 2572.4 | 3144.83 | 3488.36 | 4465.75 |
| | | AUTOMATA | 2780.04 | 3269.64 | 3860.69 | 4961.92 |
| SST5 | LSTM | FULL | 53837.27 | 53837.27 | 53837.27 | 53837.27 |
| | | RANDOM | 15305.55 | 17552.31 | 22505.57 | 34162.07 |
| | | CRAIG | 15988.2 | 18017.88 | 21071.32 | 34704.8 |
| | | AUTOMATA | 18563.96 | 21902.34 | 26904.39 | 38708.04 |
| glue-SST2 | LSTM | FULL | 8711.01 | 8711.01 | 8711.01 | 8711.01 |
| | | RANDOM | 2918.03 | 2897.04 | 3609.69 | 4175.74 |
| | | CRAIG | 2950.79 | 3095.58 | 3490.73 | 5004.83 |
| | | AUTOMATA | 3226.54 | 3272.58 | 3376.85 | 4725.78 |
| TREC6 | LSTM | FULL | 7399.83 | 7399.83 | 7399.83 | 7399.83 |
| | | RANDOM | 2198.11 | 2319.05 | 2617.33 | 3673.78 |
| | | CRAIG | 2209.83 | 2345.89 | 2757.99 | 3478.18 |
| | | AUTOMATA | 2291.19 | 2482.25 | 2593.89 | 3435.26 |
| **Search Algorithm, Scheduler** | | | **Random, ASHA** | | | |
| | | | Average Wall Clock Time in seconds(for 5 runs) | | | |
| | | Budget(%) | 1% | 5% | 10% | 30% |
| Dataset | Model | Selection Strategy | | | | |
| SST2 | LSTM | FULL | 16123.63 | 16123.63 | 16123.63 | 16123.63 |
| | | RANDOM | 4396.81 | 5623.68 | 7114.67 | 10322.79 |
| | | CRAIG | 4556.88 | 5852.1 | 7095.12 | 11590.89 |
| | | AUTOMATA | 4761.83 | 6034.1 | 7569.32 | 11893.21 |
| SST5 | LSTM | FULL | 124759.67 | 124759.67 | 124759.67 | 124759.67 |
| | | RANDOM | 35977.68 | 41598.92 | 43515.87 | 61903.21 |
| | | CRAIG | 37360.63 | 45228.69 | 45584.93 | 64019.95 |
| | | AUTOMATA | 38176.31 | 47398.21 | 48841.93 | 68019.95 |
| glue-SST2 | LSTM | FULL | 15821.32 | 15821.32 | 15821.32 | 15821.32 |
| | | RANDOM | 6423.93 | 6315.01 | 6987.43 | 8722.98 |
| | | CRAIG | 6444.7 | 6343.33 | 7254.09 | 8939.34 |
| | | AUTOMATA | 6659.99 | 6641 | 7446.55 | 9125.92 |
| TREC6 | LSTM | FULL | 10955.42 | 10955.42 | 10955.42 | 10955.42 |
| | | RANDOM | 3309.6 | 3735.43 | 4509.41 | 6408.9 |
| | | CRAIG | 3393.93 | 3877.03 | 4737.37 | 6921.3 |
| | | AUTOMATA | 3429.82 | 4091.33 | 4976.39 | 7266.64 |

Table 7: Average Wall Clock Time in seconds for SST2, SST5, glue-SST2 and TREC6 datasets for 5 runs

| Average Wall Clock Time Results | | | | | | |
|---|---|---|---|---|---|---|
| Search Algorithm, Scheduler | | | TPE, HyperBand | | | |
| | | | Average Wall Clock Time in seconds(for 3 runs) | | | |
| | | Budget(%) | 1% | 5% | 10% | 30% |
| Dataset | Model | Selection Strategy | | | | |
| CIFAR100 | ResNet18 | FULL | 269064.2 | 269064.2 | 269064.2 | 269064.2 |
| | | RANDOM | 20598 | 41254.2 | 59511.6 | 153178 |
| | | CRAIG | 22747 | 45050 | 65511 | 153178 |
| | | AUTOMATA | 41196 | 70132.14 | 92242.98 | 214449.2 |
| CIFAR10 | ResNet18 | FULL | 317823.16 | 317823.16 | 317823.16 | 317823.16 |
| | | RANDOM | 22969.8 | 46269.76 | 69131.35 | 124078.97 |
| | | CRAIG | 23365.8 | 43552.59 | 67141.09 | 129324.73 |
| | | AUTOMATA | 43226.73 | 74039.4 | 107425.74 | 199160.08 |
| SVHN | ResNet18 | FULL | 310874.12 | 310874.12 | 310874.12 | 310874.12 |
| | | RANDOM | 21569.8 | 43245.76 | 67827.13 | 123982.12 |
| | | CRAIG | 22653.8 | 42152.59 | 66341.09 | 127324.73 |
| | | AUTOMATA | 40098.53 | 71237.73 | 109462.83 | 197353.33 |
| Search Algorithm, Scheduler | | | Random, HyperBand | | | |
| | | | Average Wall Clock Time in seconds(for 3 runs) | | | |
| | | Budget(%) | 1% | 5% | 10% | 30% |
| Dataset | Model | Selection Strategy | | | | |
| CIFAR100 | ResNet18 | FULL | 389064.2 | 389064.2 | 389064.2 | 389064.2 |
| | | RANDOM | 33863.68 | 40880.12 | 53586.84 | 96832.36 |
| | | CRAIG | 25862.88 | 34680.82 | 44906.34 | 89658.28 |
| | | AUTOMATA | 64340.99 | 69496.2 | 88418.29 | 145248.5 |
| CIFAR10 | ResNet18 | FULL | 335472.16 | 335472.16 | 335472.16 | 335472.16 |
| | | RANDOM | 22294.17 | 47701.61 | 68984.38 | 130908.8 |
| | | CRAIG | 22961.73 | 44845.86 | 66611.34 | 130655.4 |
| | | AUTOMATA | 40871.88 | 74892.59 | 105245.92 | 199902.76 |
| SVHN | ResNet18 | FULL | 316482.56 | 316482.56 | 316482.56 | 316482.56 |
| | | RANDOM | 22843.2 | 42492.91 | 68341.48 | 126464.44 |
| | | CRAIG | 25473.92 | 44782.31 | 71839.31 | 128931.41 |
| | | AUTOMATA | 42648.31 | 74813.21 | 108431.51 | 198431.64 |
| Search Algorithm, Scheduler | | | TPE, ASHA | | | |
| | | | Average Wall Clock Time in seconds(for 3 runs) | | | |
| | | Budget(%) | 1% | 5% | 10% | 30% |
| Dataset | Model | Selection Strategy | | | | |
| CIFAR100 | ResNet18 | FULL | 90034.54 | 90034.54 | 90034.54 | 90034.54 |
| | | RANDOM | 23124.42 | 31468.57 | 40321.67 | 46245.31 |
| | | CRAIG | 25724.21 | 35394.69 | 43664.3 | 51325.85 |
| | | AUTOMATA | 43254.21 | 56743.81 | 66781.12 | 86154.32 |
| CIFAR10 | ResNet18 | FULL | 127519.2 | 127519.2 | 127519.2 | 127519.2 |
| | | RANDOM | 32431.3 | 39786.24 | 49160.57 | 59761.18 |
| | | CRAIG | 33775.3 | 45689.95 | 55364.38 | 65118.28 |
| | | AUTOMATA | 57719.12 | 77672.91 | 91351.23 | 99677.42 |
| SVHN | ResNet18 | FULL | 97942.74 | 97942.74 | 97942.74 | 97942.74 |
| | | RANDOM | 24739.41 | 28430.81 | 34935.91 | 38951.51 |
| | | CRAIG | 27983.46 | 29318.39 | 34131.86 | 40319.18 |
| | | AUTOMATA | 48706.85 | 52094.63 | 54761.86 | 61688.24 |
| Search Algorithm, Scheduler | | | Random, ASHA | | | |
| | | | Average Wall Clock Time in seconds(for 3 runs) | | | |
| | | Budget(%) | 1% | 5% | 10% | 30% |
| Dataset | Model | Selection Strategy | | | | |
| CIFAR100 | ResNet18 | FULL | 94534.21 | 94534.21 | 94534.21 | 94534.21 |
| | | RANDOM | 25782.16 | 34171.12 | 43752.85 | 49871.24 |
| | | CRAIG | 28426.21 | 38548.69 | 47962.74 | 55673.31 |
| | | AUTOMATA | 49142.17 | 60312.21 | 70841.64 | 90675.31 |
| CIFAR10 | ResNet18 | FULL | 136427.73 | 136427.73 | 136427.73 | 136427.73 |
| | | RANDOM | 34464.86 | 41610.94 | 50342.24 | 61435.95 |
| | | CRAIG | 34281.31 | 49263.42 | 58963.75 | 68931.64 |
| | | AUTOMATA | 58621.04 | 83747.43 | 94342 | 106154.73 |
| SVHN | ResNet18 | FULL | 105193.35 | 105193.35 | 105193.35 | 105193.35 |
| | | RANDOM | 22843.63 | 34781.82 | 41093.24 | 58471.62 |
| | | CRAIG | 25878.6 | 37143.31 | 43084.8 | 61260.56 |
| | | AUTOMATA | 45546.34 | 60543.6 | 69632.51 | 90665.63 |

Table 8: Average Wall Clock Time in seconds for CIFAR100, CIFAR10, and SVHN datasets for 3 runs

| Average Wall Clock Time Results | | | | | | |
|---|---|---|---|---|---|---|
| Search Algorithm, Scheduler | | | TPE, HyperBand | | | |
| | | | Average Wall Clock Time in seconds(for 5 runs) | | | |
| | | Budget(%) | 1% | 5% | 10% | 30% |
| Dataset | Model | Selection Strategy | | | | |
| DNA | MLP | FULL | 387.86 | 387.86 | 387.86 | 387.86 |
| | | RANDOM | 167.84 | 171.05 | 188.16 | 208.22 |
| | | CRAIG | 165.77 | 181.06 | 186.15 | 245.61 |
| | | AUTOMATA | 390.71 | 401.11 | 425.96 | 476.53 |
| SATIMAGE | MLP | FULL | 641.7 | 641.7 | 641.7 | 641.7 |
| | | RANDOM | 240.91 | 229.43 | 260.44 | 330.54 |
| | | CRAIG | 248.01 | 267.3 | 308.76 | 370.85 |
| | | AUTOMATA | 481.27 | 508.85 | 532.66 | 595.4 |
| LETTER | MLP | FULL | 1886.45 | 1886.45 | 1886.45 | 1886.45 |
| | | RANDOM | 525.65 | 577.85 | 648.37 | 941.61 |
| | | CRAIG | 569.2 | 650.46 | 785.31 | 1112.14 |
| | | AUTOMATA | 1183.86 | 1187.32 | 1296.27 | 1836.26 |
| CONNECT-4 | MLP | FULL | 8683.98 | 8683.98 | 8683.98 | 8683.98 |
| | | RANDOM | 729.13 | 1018.83 | 1181.82 | 2834.54 |
| | | CRAIG | 1170.45 | 1284.85 | 1959.46 | 3863.73 |
| | | AUTOMATA | 2209.92 | 2350.57 | 3016.15 | 4865.57 |
| Search Algorithm, Scheduler | | | Random, HyperBand | | | |
| | | | Average Wall Clock Time in seconds(for 5 runs) | | | |
| | | Budget(%) | 1% | 5% | 10% | 30% |
| Dataset | Model | Selection Strategy | | | | |
| DNA | MLP | FULL | 283.94 | 283.94 | 283.94 | 283.94 |
| | | RANDOM | 140.06 | 145.92 | 166.12 | 188.86 |
| | | CRAIG | 150.11 | 166.59 | 170.21 | 203.83 |
| | | AUTOMATA | 373.56 | 385.59 | 414.01 | 435.17 |
| SATIMAGE | MLP | FULL | 588.77 | 588.77 | 588.77 | 588.77 |
| | | RANDOM | 208.3 | 221.85 | 259.66 | 332.14 |
| | | CRAIG | 221.97 | 247.34 | 293.83 | 325.07 |
| | | AUTOMATA | 466.11 | 455.99 | 476.23 | 570.63 |
| LETTER | MLP | FULL | 1630.65 | 1630.65 | 1630.65 | 1630.65 |
| | | RANDOM | 497.58 | 480.67 | 607.6 | 862.48 |
| | | CRAIG | 502.41 | 537.59 | 642.87 | 945.89 |
| | | AUTOMATA | 1056.58 | 1195.08 | 1258.16 | 1703.97 |
| CONNECT-4 | MLP | FULL | 8683.98 | 8683.98 | 8683.98 | 8683.98 |
| | | RANDOM | 713.09 | 967.41 | 1260.49 | 2568.76 |
| | | CRAIG | 1030.85 | 1233.9 | 1604.37 | 3645.16 |
| | | AUTOMATA | 2048.01 | 2286.52 | 2633.35 | 4529.69 |
| Search Algorithm, Scheduler | | | TPE, ASHA | | | |
| | | | Average Wall Clock Time in seconds(for 5 runs) | | | |
| | | Budget(%) | 1% | 5% | 10% | 30% |
| Dataset | Model | Selection Strategy | | | | |
| DNA | MLP | FULL | 114.57 | 114.57 | 114.57 | 114.57 |
| | | RANDOM | 73.28 | 85.92 | 96.53 | 103.41 |
| | | CRAIG | 67.24 | 71.61 | 91.27 | 102.08 |
| | | AUTOMATA | 100.25 | 119.11 | 131.08 | 162.78 |
| SATIMAGE | MLP | FULL | 132.83 | 132.83 | 132.83 | 132.83 |
| | | RANDOM | 73.84 | 83.58 | 96.83 | 105.19 |
| | | CRAIG | 72.43 | 81.24 | 95.43 | 101.73 |
| | | AUTOMATA | 120.23 | 147.61 | 153.98 | 213.77 |
| LETTER | MLP | FULL | 664.02 | 664.02 | 664.02 | 664.02 |
| | | RANDOM | 187.55 | 254.85 | 312.89 | 431.98 |
| | | CRAIG | 196.36 | 278.87 | 329.65 | 437.18 |
| | | AUTOMATA | 299.47 | 358.43 | 413.17 | 686.63 |
| CONNECT-4 | MLP | FULL | 2120.13 | 2120.13 | 2120.13 | 2120.13 |
| | | RANDOM | 402.25 | 532.1 | 622.04 | 678.87 |
| | | CRAIG | 425.98 | 527.49 | 735.57 | 917.3 |
| | | AUTOMATA | 815.22 | 839.31 | 906.25 | 932.89 |
| Search Algorithm, Scheduler | | | Random, ASHA | | | |
| | | | Average Wall Clock Time in seconds(for 5 runs) | | | |
| | | Budget(%) | 1% | 5% | 10% | 30% |
| Dataset | Model | Selection Strategy | | | | |
| DNA | MLP | FULL | 127.8 | 127.8 | 127.8 | 127.8 |
| | | RANDOM | 62.95 | 75.46 | 77.83 | 101.64 |
| | | CRAIG | 75.63 | 78.5 | 83.38 | 110.16 |
| | | AUTOMATA | 92.13 | 115.35 | 121.06 | 131.58 |
| SATIMAGE | MLP | FULL | 129.88 | 129.88 | 129.88 | 129.88 |
| | | RANDOM | 74.17 | 86.29 | 91.37 | 107.92 |
| | | CRAIG | 71.13 | 83.78 | 90.84 | 102.94 |
| | | AUTOMATA | 90.69 | 115.34 | 128.61 | 204.13 |
| LETTER | MLP | FULL | 483.42 | 483.42 | 483.42 | 483.42 |
| | | RANDOM | 166.16 | 232.18 | 243.13 | 352.18 |
| | | CRAIG | 186.86 | 238.91 | 255.71 | 396.67 |
| | | AUTOMATA | 295.1 | 320.05 | 415.75 | 711.48 |
| CONNECT-4 | MLP | FULL | 1810.66 | 1810.66 | 1810.66 | 1810.66 |
| | | RANDOM | 380.02 | 603.7 | 655.0 | 840.8 |
| | | CRAIG | 426.89 | 654.99 | 691.46 | 914.48 |
| | | AUTOMATA | 660.01 | 736.67 | 793.31 | 1109.96 |

Table 9: Average Wall Clock Time in seconds for DNA, SATIMAGE, LETTER, and CONNECT-4 datasets for 5 runs

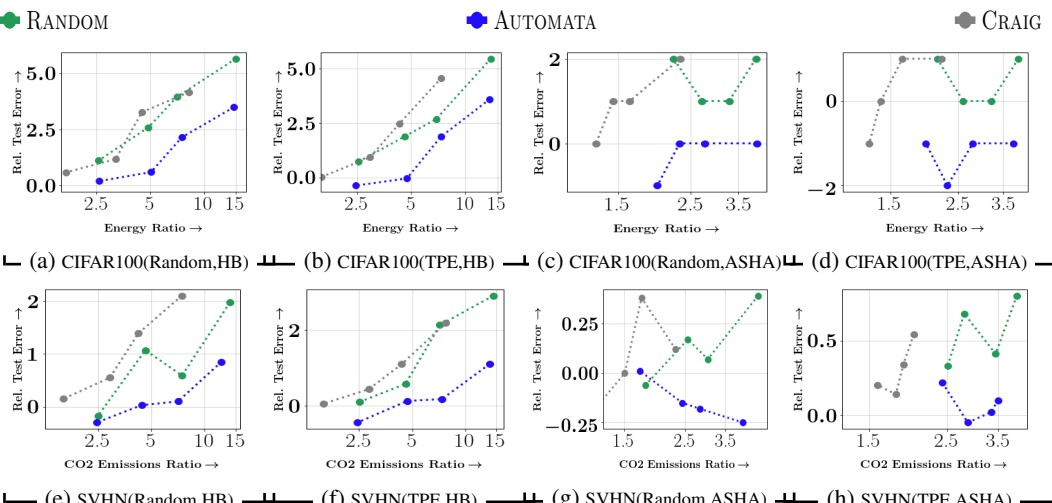

Figure 12: Comparison of performance of AUTOMATA with baselines(RANDOM, CRAIG, FULL) for Hyper-parameter tuning. In sub-figures (a-d), we present energy ratio *vs.* relative test error (in %), compared to Full data tuning for different methods on CIFAR100 dataset. In sub-figures (e-h), we present co2 emissions ratio *vs.* relative test error (in %), compared to Full data tuning for different methods on SVHN dataset. On each scatter plot, smaller subsets appear on the right, and larger ones appear on the left. *The scatter plots show that* AUTOMATA *achieves the best energy savings and CO2 reductions, thereby achieving the best efficiency vs. performance tradeoff in almost every case. (**Bottom-right corner of each plot indicates the best efficiency vs. performance tradeoff region**).*