# OpenReview forum: "AUTOMATA: Gradient Based Data Subset Selection for Compute-Efficient Hyper-parameter Tuning"
_NeurIPS.cc/2022/Conference — NeurIPS 2022 Accept_

### Official Review · Reviewer_HuPX · 2022-07-01

**Rating:** 6
**Confidence:** 3
**Soundness:** 2 fair
**Presentation:** 2 fair
**Contribution:** 3 good

**Summary:**

The paper presents AUTOMATA, a framework for hyper parameter tuning that relies on three main components for accelerating the search for the optimal hyper-parameter configuration. Automata was developed with the goal of accelerating the search for the optimal configuration in order to minimize the cost and environmental impact of the hyper-parameter tuning process. The key mechanism described by the authors as the main contributor to the savings and speed-ups achieved by Automata is DSS, a data subset selection approach that aims to find a subset of data with which to evaluate each configuration such that: i) it is faster to evaluate the quality of a configuration; ii) the performance order of the evaluated configurations when evaluated with the full training set is preserved.


**Questions:**

**a)** The expression "retaining the ordering of hyper-parameters" (line 73) is not explained well enough. From my understanding, what it means is that given that Automata uses less data to evaluate a specific configuration, you want to guarantee that if configuration A is worse than configuration B (when evaluated with a data subset selected by Automata), configuration A is actually worse than configuration B when evaluated with the full dataset. Is this correct? If yes, I could only understand this half-way through the paper and if you could clarify this earlier on, the paper would read much better.

**b)** Figure 2 could be improved and further leveraged to aid with the explanation of the data subset selection procedure described in Section 2.3.1. For instance, why are there multiple DSS training loops? Do you evaluate multiple hyper parameter configurations at once? If yes, you could perhaps add this information to each of the input arrows to the DSS training loop component. It would also help if you added an example of the "normal" subset selection mechanism to visually compare it with the per-batch method.

**c)** What is the meaning of k in equation 1) and in line 186? Is it the same k of line 209? If so, then how/where are coresets used by the data subset selection mechanism?

**d)** Why do you use only MNIST to evaluate whether GSS retains the order of configurations if for the remaining evaluation you never use MNIST again? Since you have results for FULL for each pair of dataset and framework configuration (i.e., scheduler and hyper parameter tuning approach), why not evaluate the order retention for those cases as well? Moreover, since you claim Automata has better performance on larger datasets and models (line 351), it would be very interesting to see if the configuration order is retained in these cases.

**e)** Can you clarify what you mean by relative test error?

**f)** How do DSS and Grad-Match differ? If they differ, why is Grad-Match not a baseline? If they do not differ, this should be clarified in the submission.

**Observations:**
- checklist question - Did you include the total amount of compute and the type of resources used (e.g., type 570 of GPUs, internal cluster, or cloud provider)? --- I could not find this information in section F.4
- Supplemental material, figure 10: it seems like the x-axis in the figure are not correctly labelled.


**Limitations:**

Limitations regarding any potential negative societal impact of the work were addressed.

**Strengths And Weaknesses:**

**Originality:** the tasks and the methods are not new but the work provides a combination of well-known techniques which, as far as I know, is novel. It is not clear how DSS differs from Grad-Match [23].

**Quality:** The evaluation could be improved by providing more experiments on the order of the hyper-parameters. If DSS differs from Grad-Match, why is Grad-Match not a baseline?

**Clarity:** Overall, the paper is clear and well written. Some parts could be improved. First, if Automata is a framework, then more emphasis should be given to its components and to its use. For example, if I wanted to use Automata for my work: i) which components could I reuse? ii) which components would I need to replace? iii) what should I be careful about, for instance regarding integration of the 3 main components (search algorithm, DSS, and scheduler)? Second, the meaning of "retaining the ordering of hyper-parameters" is not clearly explained anywhere in the paper and when the authors mention that final accuracy is not their main concern (line 74) confuses the reader, since in the hyper parameter tuning context, improving accuracy is a primary concern. Third, it was not clear to me if variable 'k' was the same throughout section 3 nor what its meaning was. Figure 2 currently does not contribute much to understanding the paper. Suggestions on how to improve this are below. Finally, section 3.2 would read much better if there were more paragraphs, perhaps split per domain, like in section 3.4 (text datasets, image datasets, tabular datasets).

**Significance:** the idea of having a framework that contains different search algorithms, data subset selection strategies, and schedulers that can be mixed and matched to speed-up the hyper-parameter tuning for different applications and domains is interesting and I can see it being used by practitioners/researchers.

---

> ### Author Response · Authors · 2022-08-02
> **Response to Reviewer HuPX**
>
> **Q1. Is DSS different from GradMatch?**
>
> We use the Per-Batch version of GradMatch as DSS. We mention this clearly in the submitted rebuttal version of the paper.
>
> **Q2. More experiments on the ordering of hyper-parameters**
>
> |            |                 | Spearman Rank Correlation Values                | | |
> |---------|-----------------|--------------------------------------------------|---|---|
> | Dataset | Subset Fraction | AUTOMATA                         | CRAIG | Random  |
> | CIFAR10 | 1%              | 0.621                            | 0.543 | 0.392   |
> | CIFAR10 | 5%              | 0.683                            | 0.592 | 0.431   |
> | CIFAR10 | 10%             | 0.713                            | 0.612 | 0.464   |
> | TREC6   | 1%              | 0.564                            | 0.512 | 0.487   |
> | TREC6   | 5%              | 0.704                            | 0.682 | 0.664   |
> | TREC6   | 10%             | 0.741                            | 0.708 | 0.675   |
>
> We present additional results about hyper-parameter ordering on CIFAR10 and TREC6 datasets in the above table. For both CIFAR10 and TREC6, we empirically observe that AUTOMATA retains the hyper-parameter ordering better than other subset selection baselines. We have added Figure 1b to the rebuttal version of the paper showing the hyper-parameter ordering retention capability (spearman rank correlation values) of different subset selection strategies for CIFAR10 and TREC6 datasets.
>
> **Q3. Bring connections to hyper-parameter ordering retention early on**
>
> Thanks for your suggestion. We have also added the corresponding explanations of the hyper-parameter order retention early in the Contributions section and while discussing the subset-based configuration evaluation, i.e., line nos 150-156. Finally, we added Figure 1b to the rebuttal version of the paper showing the hyper-parameter ordering retention capability (spearman rank correlation values) of different subset selection strategies for CIFAR10 and TREC6 datasets.
>
>  **Q4. Improve Figure 2 to explain the pipeline better. Also, visualize the difference between normal and per-batch subset selection**
>
> We update Figure 2 according to your suggestion in the rebuttal version of the paper. Yes, in practice, we evaluate multiple hyper-parameter configurations at once. We present a figure visualizing the difference between per-sample (“normal”) subset selection and per-batch subset selection in Figure 5 of the Appendix.
>
> **Q5. what is the meaning of k in equation 1) and in line 186? Is it the same k of line 209? If so, then how/where are coresets used by the data subset selection mechanism?**
>
> Yes, k in lines 186 and 209 are the same and represent the subset's size or the number of data samples that need to be selected. In line 186, we define the number of the mini-batches that need to be selected using per batch selection as the ratio of the size of the subset and the batch size. In line 209, we decide the number of warm start epochs based on the subset size so that the overall time for training does not increase due to the warm starting of the model. For each model, we use GradMatch for selecting coresets, and the selected coresets are used for training the model for the following R epochs as defined in the paper.
>
> **Q6. Can you clarify what you mean by relative test error?**
>
> Relative test error is the difference between the Test accuracy achieved using the best hyper-parameter configuration selected by hyper-parameter tuning using a data subset selection strategy for configuration evaluation and the test accuracy achieved using the best hyper-parameter configuration selected by hyper-parameter tuning using Full data for configuration evaluation.
>
> **Q7. How do DSS and Grad-Match differ? If they differ, why is Grad-Match not a baseline? If they do not differ, this should be clarified in the submission.**
>
> We use the per-batch version of GradMatch as DSS. We clarified this in the rebuttal version as well.
>
> **Q8. Checklist question - Did you include the total amount of compute and the type of resources used (e.g., type 570 of GPUs, internal cluster, or cloud provider)? --- I could not find this information in section F.4**
>
> We apologize for the confusion caused by our reference to a wrong subsection. We have added the information on GPUs in section G.1 of the Appendix and updated the checklist accordingly.
>
> **Q9. Supplemental material, figure 10: it seems like the x-axis in the figure is not correctly labeled.**
>
> Thanks for pointing out the mistake. We apologize for the typo in the x-axis labels and the confusion it caused. We corrected the x-axis names in the rebuttal version of the paper.

---

> > ### Comment · Reviewer_HuPX · 2022-08-08
> > **Thank you for the clarifications**
> >
> > I would like to thank the authors for the clarifications. I have no further questions.

---

### Official Review · Reviewer_9uDV · 2022-07-09

**Rating:** 6
**Confidence:** 4
**Soundness:** 3 good
**Presentation:** 4 excellent
**Contribution:** 2 fair

**Summary:**

This work proposed a framework to speed up HPO by gradient based dataset subset selection. The methodology contribution lies on the gradient based per batch subset selection using OMP. It seems to be a modification of  GARD-MATCH [23]. The authors provided extensive empirical results on different HPO search algorithms and HPO schedule algorithms, and in combination of different core set selection methods. The results suggested that one could speed up HPO significantly with the framework while keeping the final solution quality (mild decreasing).

**Questions:**

* It seems the results can be quite different depending on the HPO search algorithm and scheduler. Why is it the case and could the authors provide some insights? Does this mean I would better not to use the proposed method for certain HPO optimizer?

* The author mentioned “Similar to GRAD-MATCH, we use a greedy algorithm called orthogonal matching pursuit (OMP)”, what is then the difference?

* Figure 3 doesn't show any error bar but there are 3 or 5 repetitions. Is there some particular reason for it?

* [Minor] Line 190, the summation over all the batches with subscript $i$ should be with something else as $i$ is already the hyperparameter configuration.


**Limitations:**

The authors have mentioned that choosing a subset size in advance can be difficult and it is hard to anticipate the loss of performance. I would also add how to make it more robust across different HPO optimizers.

**Strengths And Weaknesses:**

The paper’s empirical work is very well done. It contains clear experiments setting, abundant real world examples, thoughtful baselines, detailed ablation studies and interesting analysis. The results are also encouraging. I like the presentation, the paper is well organized and easy to follow.

On the weakness side, the methodology contribution seems limited and the main merits of the paper are the empirical results. Also, the trade off between speed up and performance loss is hard to quantify in advance and it will be hard to choose a subset size in practice without trying a few options. In the end, the results seem sensitive to the HPO optimizers (the performance loss could differ significantly).

---

> ### Author Response · Authors · 2022-08-02
> **Response to Reviewer 9uDV**
>
> **Q1. It seems the results can be quite different depending on the HPO search algorithm and scheduler. Why is it the case, and could the authors provide some insights? Does this mean I would better not use the proposed method for a certain HPO optimizer?**
>
> AUTOMATA can be used in conjunction with any hyper-parameter search algorithm and scheduler. It should be noted that the speedups obtained using AUTOMATA are heavily dependent upon the scheduler. Some schedulers, such as ASHA, are highly efficient and can effectively discard poor hyper-parameter configurations early, thus reducing the advantages caused by using data subsets. Therefore, the speedup achieved by AUTOMATA may be lower if a more efficient scheduler is used rather than a less efficient one. Moreover, our empirical results confirm the same results, with AUTOMATA achieving speedups of approximately 3x when using ASHA as a scheduler and around 10x-15x when using Hyperband. Nevertheless, we recommend using AUTOMATA with any scheduler since it can still reduce tuning time and thus power consumption. We clarify this in the rebuttal version of the paper as well.
>
> **Q2. The author mentioned, “Similar to GRAD-MATCH, we use a greedy algorithm called orthogonal matching pursuit (OMP)” what is then the difference?**
>
> We apologize for the confusion caused by this statement. We use the per-batch version of GradMatch as the DSS strategy as it was shown to be the most efficient one by the authors of the GradMatch work. We modified this statement in the rebuttal version of the paper.
>
> **Q3. Figure 3 doesn't show any error bar, but there are 3 or 5 repetitions. Is there some particular reason for it?**
>
> We presented standard deviations in Table 4,5,6 of the Appendix. Standard deviation results show that AUTOMATA achieves results with low variance consistently compared to other subset selection baselines. In the paper's rebuttal version, we plotted the error bars.
>
> **Q4. [Minor] Line 190, the summation over all the batches with subscript i should be with something else as i is already the hyperparameter configuration.**
>
> Thanks for pointing out the typo. We have corrected it in the submitted rebuttal version of the paper.

---

> > ### Comment · Reviewer_9uDV · 2022-08-08
> > **Reply to authors' response**
> >
> > I'd like to thank the authors for their reply. I don't have major concerns on the methodology and the results. I'd like to maintain my score mostly because the weaknesses that have been mentioned.

---

### Official Review · Reviewer_Grjc · 2022-07-12

**Rating:** 4
**Confidence:** 4
**Soundness:** 2 fair
**Presentation:** 2 fair
**Contribution:** 2 fair

**Summary:**

This paper presents AUTOMATA, a gradient-based subset selection framework for hyper-parameter tuning. The authors evaluate the framework using various datasets and models. The experimental results show that the proposed framework can improve the speedup of parameter tuning significantly.

**Questions:**

- What are the hyper-parameter scope used in the experiments?
- The experiment only compares with CRAIG. How about other approaches?
- It will be helpful for the authors to explain why the ASHA or Hyperband are used as scheduling algorithm? Are these two the most representative scheduling algorithms or the most commonly used ones?
- Does this approach apply to more complex datasets, such as ImageNet?
- How about showing the training wall clock time?
- Have you evaluate the cases where the subsets are chosen at different number of epochs, e.g., 10, 40, etc?


**Limitations:**

The authors have addressed the limitations of their works.

**Strengths And Weaknesses:**

The method proposed by the authors are very interesting.

Originality:
The novelty of this work limited. The design is more about a combination of some existing techniques.

Quality:
- The writing can be improved. There is no explanation and justification on some design decisions. Please refer to Questions section for more details.

Clarity:
- It will be helpful to provide more justification to the design. Why those three components are crucial?
- The main text should be self-contained.
- In Section 2.3.1, there is no theoretical proof or experiment to support idea of gradient selection.
- Better to informative component names instead of Component-1, 2, and 3.

Significance:
- This speedup-accuracy trade-off can be useful for other follow up works.

---

> ### Author Response · Authors · 2022-08-02
> **Response to Reviewer Grjc**
>
> **Q1. Justification for the design. Why those three components are crucial?**
>
> We did explain the importance of the different components of AUTOMATA in section 2 of the main paper. Intuitively, given a search space, we need to find the best hyper-parameter configuration through tuning. Since the number of possible configuration evaluations is limited, SOTA hyper-parameter search algorithms need to be used to sample better-performing configurations from search space for evaluation effectively. Further, to speed up each configuration evaluation, we adopt subset selection strategies for faster model training. Finally, we can gain further efficiency by discarding poor-performing configurations early instead of training them to the end. Hence, schedulers form a final component of our framework.
>
> **Q2. Self-contained main text.**
>
> We worked on making our paper as clear as possible by making necessary changes during the rebuttal.
>
> **Q3. No theoretical proof or experiment to support gradient selection.**
>
>
> |            |                 | Spearman Rank Correlation Values                | | |
> |---------|-----------------|--------------------------------------------------|---|---|
> | Dataset | Subset Fraction | AUTOMATA                         | CRAIG | Random  |
> | CIFAR10 | 1%              | 0.621                            | 0.543 | 0.392   |
> | CIFAR10 | 5%              | 0.683                            | 0.592 | 0.431   |
> | CIFAR10 | 10%             | 0.713                            | 0.612 | 0.464   |
> | TREC6   | 1%              | 0.564                            | 0.512 | 0.487   |
> | TREC6   | 5%              | 0.704                            | 0.682 | 0.664   |
> | TREC6   | 10%             | 0.741                            | 0.708 | 0.675   |
>
> We use \textsc{GradMatch}, a gradient-based subset selection strategy in \model\ because \textsc{GradMatch} can preserve the original ordering of hyper-parameters better than \textsc{Random} and other baselines even when using small subset sizes, as shown by the Spearman ranking correlation values in the above Figure. Hence, the critical advantage of AUTOMATA is that we can achieve speedups while retaining the hyper-parameter ordering allowing us to find the best hyperparameters. We also mention the empirical motivation behind using gradient-based subset selection in lines 150-153.
>
> **Q4. Informative component names.**
>
> Thanks for your suggestion. We presented informative component names in Section 2 of the paper.
>
> **Q5. Hyper-parameter scope used in the experiments?**
>
> We present details on the hyper-parameter search spaces considered for our experiments in the Experimental section of the main paper and section G.4 of the Appendix.
>
> **Q6. Comparison only with CRAIG**
>
> Our motivation to consider CRAIG as a baseline is to showcase the effectiveness of the subset selection approach adopted by AUTOMATA, i.e., GradMatch, over a SOTA gradient-based subset selection strategy. Further, we compare with Random subset selection, which is most efficient in tuning time. Finally, we do not consider additional subset selection approaches like GLISTER as baselines due to compute restrictions.
>
> **Q7. Why ASHA or Hyperband as schedulers?**
>
> As mentioned in the paper, we developed AUTOMATA using the RayTune library, which contains ASHA and Hyperband as schedulers. Therefore, we used them as two representative scheduling algorithms, which are also commonly used.
>
> **Q8. Application to more complex datasets like ImageNet?**
>
> Our method can be applied to more complex datasets like ImageNet. However, we need efficient memory management to store gradients of around a million data samples in ImageNet.
>
> **Q9. Training wall clock time?**
>
> We added the training wall clock times to Table 7,8,9 of the revised Appendix.
>
> **Q10. Performance when subsets are chosen at different epochs?**
>
>
> | Epoch Interval (R)  | Test Accuracy | Time Taken |
> |----|---------------|------------|
> | 1  | 93.86 %       | 17.389     |
> | 5  | 94.06 %       | 10.7223    |
> | 10 | 93.94 %       | 9.9945     |
> | 20 | 93.92 %       | 9.656      |
> | 40 | 93.02 %       | 9.1693     |
>
> We performed an ablation study to find the best epoch interval value($R$) for data subset selection. To achieve this, we experimented on the CIFAR10 dataset with the ResNet18 model and the same hyper-parameter search space used in the rest of the CIFAR10 experiments using TPE and HyperBand as a scheduler for $R$ values of $1, 5, 10, 20, 40$ respectively. In the table above, we present the accuracies achieved by AUTOMATA at a 30% subset for different R values. The results show that AUTOMATA achieved the best performance vs. efficiency tradeoff when $R = 20$. Using $R=40$, a lower efficiency gain could be achieved at the expense of a greater performance loss. Using $R$ values of 1, 5, and 10, we achieve similar performance to $R=20$ with a significant increase in tuning time. Therefore, our experiments used an epoch interval value of $R=20$.

---

### Official Review · Reviewer_Mepf · 2022-07-12

**Rating:** 6
**Confidence:** 4
**Soundness:** 3 good
**Presentation:** 2 fair
**Contribution:** 3 good

**Summary:**

This paper evaluates a gradient-based subset selection method in hyperparameter optimization and shows that the method can speed up the hyperparameter optimization process without significant performance deterioration compared to when using the entire dataset. The proposed method, AUTOMATA, consists of three components: hyperparameter search algorithm, data subset selection method, and hyperparameter scheduler. Each component of the proposed method can be selected from existing algorithms. The effectiveness of AUTOMATA is experimentally verified on datasets of text, computer vision, and tabular domain. The performance of  AUTOMATA is experimentally verified.

**Questions:**

* The effectiveness of the combination of data subset selection and hyperparameter scheduling is not so surprising because these techniques attempt to speed up model training from different aspects. It would be better to clarify the difficulty of incorporating the data subset selection into hyperparameter optimization.

* I think that the value of the epoch interval for subset selection impacts the computational efficiency and accuracy in AUTOMATA. What is the impact of the epoch interval setting on the performance?

* The hyperparameter search space treated in this paper seems to be not so large. It would be great to show the experimental results in large-scale search spaces. The NAS-Bench or HPO-Bench would be possible choices.

* In Figure 3, the plot of "FULL" (red-colored) cannot be found.

* The error bar should be plotted in Figure 3.

**Limitations:**

The limitations and potential negative societal impact are addressed.


**Strengths And Weaknesses:**

* Strengths: The effectiveness of the data subset selection method in hyperparameter optimization is experimentally validated, and the proposed AUTOMATA succeeds in speedups of 3 to 30 times for several datasets.

* Weaknesses: The approach is somewhat straightforward. The novelty of this paper is using the gradient-based data subset selection method in hyperparameter optimization. It is quite natural that subset selection methods are also useful in hyperparameter optimization because existing data subset selection methods succeed in the cost reduction of a single training process.

---

> ### Author Response · Authors · 2022-08-02
> **Response to Reviewer Mepf (2/2)**
>
> **Q4. The hyperparameter search space treated in this paper seems to be not so large. It would be great to show the experimental results in large-scale search spaces. The NAS-Bench or HPO-Bench would be possible choices.**
>
> In Figure 4 of the main paper, we present experimental results with a large search space (110 configurations) on the CIFAR10 dataset. Given the resource and computational constraints, we did not experiment with even larger search spaces. We focused primarily on large-scale vision and text datasets using deep models, where subset selection is paramount, and there are substantial advantages to utilizing subset selection. Additionally, we did not consider HPO-Bench because most of the experiments in HPO-Bench are conducted on tabular datasets, where subset selection is less likely to be necessary due to quick individual model training times. Additionally, we utilize GradMatch as a strategy for selecting subsets, a method that requires a differentiable model and is incompatible with commonly used tabular classification models such as Boosted Trees.
>
> **Q5. In Figure 3, the plot of "FULL" (red-colored) cannot be found.**
>
> We apologize for any confusion caused by the word "FULL" in the legend. All results in our work are plotted relative to the FULL, which is why FULL is not seen. To prevent confusion, we removed the “FULL” legend in the plots of the rebuttal version of the paper.
>
> **Q6. The error bar should be plotted in Figure 3.**
>
> We presented standard deviations in Table 4,5,6 of the Appendix. Standard deviation results show that AUTOMATA achieves results with low variance consistently compared to other subset selection baselines. In the paper's rebuttal version, we plotted the error bars.

---

> > ### Comment · Reviewer_Mepf · 2022-08-09
> > **Thank you for the response**
> >
> > Thank you for the detailed response. Basically, the author's response is convincing me. The difficulty of applying the data subset selection to hyperparameter optimization was made clear by the discussion based on the Spearman rank correlation. I appreciate that the authors added Figure 1 (b) regarding this result to the revised paper, which will be useful to understand the motivation and contribution of this study.
> >
> > Also, thank you for showing the additional experimental results on the impact of the epoch interval. It is also convincing me.
> >
> > To summarize, I am satisfied with the author's response and raised my score to the positive side.

---

> ### Author Response · Authors · 2022-08-02
> **Response to Reviewer Mepf (1/2)**
>
>
> **Q1. The approach is somewhat straightforward. The novelty of this paper is using the gradient-based data subset selection method in hyperparameter optimization. It is pretty natural that subset selection methods are also helpful in hyperparameter optimization because existing data subset selection methods succeed in the cost reduction of a single training process.**
>
> We believe applying any subset selection approach for hyper-parameter tuning is not straightforward. We need to preserve the original ordering of hyper-parameters even when the models are trained on small subsets. To this extent, we also present insights on the effectiveness of different data subset selection strategies in retaining hyper-parameter ordering below:
>
> |            |                 | Spearman Rank Correlation Values                | | |
> |---------|-----------------|--------------------------------------------------|---|---|
> | Dataset | Subset Fraction | AUTOMATA                         | CRAIG | Random  |
> | CIFAR10 | 1%              | 0.621                            | 0.543 | 0.392   |
> | CIFAR10 | 5%              | 0.683                            | 0.592 | 0.431   |
> | CIFAR10 | 10%             | 0.713                            | 0.612 | 0.464   |
> | TREC6   | 1%              | 0.564                            | 0.512 | 0.487   |
> | TREC6   | 5%              | 0.704                            | 0.682 | 0.664   |
> | TREC6   | 10%             | 0.741                            | 0.708 | 0.675   |
>
> The above table shows spearman rank correlation values on CIFAR10 and TREC6 datasets; the higher the rank correlation value, the better the hyper-parameter ordering retention capability of subset selection strategies. For both CIFAR10 and TREC6, we empirically observe that AUTOMATA retains the hyper-parameter ordering better than other subset selection baselines. The results show that some subset selection strategies could preserve the original ordering of the hyper-parameters better than others.
>
> Further, In this work, we perform a detailed empirical study of the effect of intelligent subset selection methods for hyper-parameter tuning when used with different search algorithms and schedulers on multiple datasets from different domains, which we strongly believe to be a valuable contribution. Finally, as we stated in our conclusion, we hope our work encourages the community to use subset selection techniques to increase hyper-parameter tuning speed and promote further research on efficient subset selection approaches.
>
> **Q2. The effectiveness of the combination of data subset selection and hyperparameter scheduling is not surprising because these techniques attempt to speed up model training from different aspects. It would be better to clarify the difficulty of incorporating the data subset selection into hyperparameter optimization.**
>
> We do discuss the difficulties of incorporating subset selection strategies into hyper-parameter tuning. One, we showed insights on the effectiveness of different data subset selection strategies in retaining hyper-parameter ordering. Our experiments on hyper-parameter ordering retention highlight that not all subset selection strategies are suitable for hyperparameter tuning, even though they are good for individual model training. We also present detailed ablation studies showing what hyperparameters should be used for GradMatch to get maximum hyper-parameter tuning performance with different search algorithms and schedulers.
>
> **Q3. I think that the value of the epoch interval for subset selection impacts AUTOMATA's computational efficiency and accuracy. What is the impact of the epoch interval setting on the performance?**
>
> | Epoch Interval (R)  | Test Accuracy | Time Taken |
> |----|---------------|------------|
> | 1  | 93.86 %       | 17.389     |
> | 5  | 94.06 %       | 10.7223    |
> | 10 | 93.94 %       | 9.9945     |
> | 20 | 93.92 %       | 9.656      |
> | 40 | 93.02 %       | 9.1693     |
>
> We ran an ablation study to find the best epoch interval ($R$) for data subset selection. To achieve this, we experimented on the CIFAR10 dataset with the ResNet18 model and the same hyper-parameter search space used in the rest of the CIFAR10 experiments using TPE and HyperBand as a scheduler for $R$ values of $1, 5, 10, 20, 40$ respectively. We present the accuracies achieved by AUTOMATA at a 30% subset for different values of R in the above table. The results show that AUTOMATA achieved the best performance vs. efficiency tradeoff when $R = 20$. Using $R=40$, a lower efficiency gain could be achieved at the expense of a greater performance loss. Using $R$ values of 1, 5, and 10, we achieve similar performance to $R=20$ with a significant increase in tuning time. Therefore, our experiments used an epoch interval value of $R=20$.
>
> We have also added the ablation study on epoch interval in Appendix G.2.3 of the rebuttal version. We also refer to the ablation study in the main paper on lines 270-272.

---

### Meta-Review · Area_Chair_MAuH · 2022-08-27

**Recommendation:** Accept
**Confidence:** Certain

**Metareview:**

This paper proposes AUTOMATA, an approach that uses GradMatch to select subsets of data in order to accelerate hyperparameter tuning. The reviewers all found the approach to be practical and empirically effective. There were concerns about the robustness to different subset sizes, particularly across different datasets, but the authors demonstrated that AUTOMATA works well across a number of settings during the rebuttal period. The remaining criticism largely revolves around the novelty of the approach, but the majority of the reviewers believe that this is a useful application of gradient-based subset selection.


**Award:**

No

---

### Decision · Program_Chairs · 2022-09-14

Accept